# Clinical phenotypes in acute and chronic infarction explained through human ventricular electromechanical modelling and simulations

Xin Zhou[1†], Zhinuo Jenny Wang[1*†], Julia Camps[1], Jakub Tomek[2], Alfonso Santiago[3,4], Adria Quintanas[3], Mariano Vazquez[3,4], Marmar Vaseghi[5,6], Blanca Rodriguez[1]

[1]Department of Computer Science, University of Oxford, Oxford, United Kingdom; [2]Department of Physiology, Anatomy and Genetics, University of Oxford, Oxford, United Kingdom; [3]Department of Computer Applications in Science and Engineering, Barcelona Supercomputing Centre (BSC), Barcelona, Spain; [4]ELEM Biotech, Barcelona, Spain; [5]UCLA Cardiac Arrhythmia Center, University of California, Los Angeles, Los Angeles, United States; [6]Neurocardiology Research Center of Excellence, University of California, Los Angeles, Los Angeles, United States

**\*For correspondence:**
jenny.wang@cs.ox.ac.uk

[†]These authors contributed equally to this work

**Competing interest:** The authors declare that no competing interests exist.

## eLife Assessment

This computational study integrates detailed electrophysiology and mechanical contraction predictions, which are often modeled separately. The findings of this **important** work are that abnormal ECGs that are associated with higher risk of sudden cardiac death are predicted to have almost no relationship with left ventricular ejection fraction, which is conventionally used as a risk factor for arrhythmia. The conclusions are based on **compelling** evidence for the need of incorporating additional risk factors for assessing post-myocardial infarction patients.

**Abstract** Sudden death after myocardial infarction (MI) is associated with electrophysiological heterogeneities and ionic current remodelling. Low ejection fraction (EF) is used in risk stratification, but its mechanistic links with pro-arrhythmic heterogeneities are unknown. We aim to provide mechanistic explanations of clinical phenotypes in acute and chronic MI, from ionic current remodelling to ECG and EF, using human electromechanical modelling and simulation to augment experimental and clinical investigations. A human ventricular electromechanical modelling and simulation framework is constructed and validated with rich experimental and clinical datasets, incorporating varying degrees of ionic current remodelling as reported in literature. In acute MI, T-wave inversion and Brugada phenocopy were explained by conduction abnormality and local action potential prolongation in the border zone. In chronic MI, upright tall T-waves highlight large repolarisation dispersion between the border and remote zones, which promoted ectopic propagation at fast pacing. Post-MI EF at resting heart rate was not sensitive to the extent of repolarisation heterogeneity and the risk of repolarisation abnormalities at fast pacing. T-wave and QT abnormalities are better indicators of repolarisation heterogeneities than EF in post-MI.

## Introduction

Sudden cardiac death in post-myocardial infarction (post-MI) patients is due to lethal arrhythmias in 50% of cases at both acute and chronic infarct stages (*Vazquez et al., 2009*; *Elayi et al., 2017*). Risk stratification is currently based on low left ventricular ejection fraction (LVEF) (*Solomon et al., 2005*) to identify patients who need the implantation of defibrillator device. However, only a very small subset of patients that suffer from sudden cardiac death are identified by low LVEF, and a significant number of sudden deaths occur in patients with relatively preserved LVEF (*Vaduganathan et al., 2018*). The mechanistic link between electrophysiological heterogeneities, ECG and LVEF phenotypes that underpin arrhythmic events is not clear. A precision medicine approach using computational modelling and simulations could help to elucidate disease mechanisms and provide an in silico alternative for therapy evaluation and risk stratification. However, a key hurdle in clinical adoption of such in silico tools is providing proof of model credibility through validation studies. In this study, we set out to achieve the dual aim of demonstrating a multi-scale approach for model validation and to identify important electrophysiological mechanisms for post-MI risk stratifications.

Various electrocardiogram (ECG) characteristics were suggested by clinical studies to be relevant to the arrhythmic risk of post-MI patients. Longer QT intervals have been associated with increased mortality as well as ventricular tachycardia and fibrillation for both the acute and chronic MI (*Ahnve, 1985*; *Oikarinen et al., 1998*). However, QT prolongation in single leads can be a reflection of reduced global repolarisation reserve, while the regional heterogeneity of repolarisation in different post-MI zones can be more crucial (*Cluitmans et al., 2021*) than the global reserve for the development of re-entrant waves. QT dispersion between 12-lead ECGs was proposed as a potential marker for electrophysiological heterogeneity *Spargias et al., 1999*; however, it was found to have good specificity but low sensitivity (*Spargias et al., 1999*). Prolonged T-peak to T-end interval (Tpe; *Shenthar et al., 2015*) and increased incidence of T-wave alternans (*Bloomfield et al., 2004*) were also found to be useful risk predictors in post-MI patients. Other ECG metrics, such as QRST integral and spatial ventricular gradients have also shown some promise in improving SCD prediction (*Waks et al., 2016*). However, despite the promising outcomes of those clinical studies, the ECG-based risk predictors are not widely applied in the clinical evaluation of the need for implantable cardioverter-defibrillators (*Al-Khatib et al., 2018*).

A key factor that hinders the clinical utility of ECG biomarkers is the large variability in post-MI phenotypes, their progression from acute to chronic, and between different patients. Variability in QT prolongation (*Ahnve, 1985*) and post-MI ECG morphologies constrain the use of single biomarker thresholds as predictors. Furthermore, an important limitation of non-invasive ECG biomarkers is their inability to provide direct measurements of regional electrophysiological heterogeneity caused by scar and ionic current remodelling, which is critical for the arrhythmic substrate.

Many experimental studies have investigated the underlying mechanisms of increased repolarisation dispersion in post-MI patients. Specifically, repolarisation dispersion increases after MI due to ionic differences between the border zone (BZ) surrounding the scar and the remote zone (RZ) myocardium (*Mendonca Costa et al., 2018*). The BZ at the acute MI exhibits reduced sodium current ($I_{Na}$), L-type calcium current ($I_{CaL}$) (*Rajesh and Boyden Penelope, 1995*), and rapid and slow delayed rectifier potassium currents ($I_{Kr}$ and $I_{Ks}$) (*Jiang et al., 2000*), as well as enhanced CaMKII activity (*Hund et al., 2008*) and gap junction redistribution (*Yao et al., 2003*). After a period of scar healing, ionic current remodelling may partially recover (*Ursell et al., 1985*), but increased late sodium current ($I_{NaL}$), calcium-activated potassium and chloride currents ($I_{KCa}$ and $I_{ClCa}$) were observed in the chronic BZ and RZ (*Hegyi et al., 2018*). Post-MI ionic current remodelling can cause conduction abnormalities, repolarisation abnormalities, as well as variable alterations in the action potential duration (APD), which act as substrates of ventricular tachycardia (*Pogwizd et al., 1992*). It is, however, unclear how various degrees of repolarisation dispersion affect ECG biomarkers and LVEF used for risk stratification. Therefore, further studies are needed to bridge the gap between cellular electrophysiological characteristics and variable patient phenotypes post-MI. Furthermore, the wealth of cellular, tissue, and ECG data described in the literature provides an excellent multi-scale dataset for model validation. Comparisons of simulated action potential and ECG biomarkers with experimental and clinical data under acute and chronic MI conditions help to improve the credibility of the computational model.

Human-based modelling and simulations of ventricular electrophysiology and mechanics have demonstrated accurate prediction of post-MI arrhythmic risk and myocardial stress and strain (*Arevalo*

*et al., 2016*; *Sack et al., 2016*). However, previous work concentrated on either electrophysiology or mechanics, while the crosstalk between the two, and the relationship between LVEF and arrhythmic risk, is yet to be investigated.

The main goal of this study is, therefore, to quantify the contribution of varying degrees of ionic current remodelling to the phenotypic variability in ECG and LVEF biomarkers observed in acute and chronic post-MI patients, and thereby create and validate models of post-MI electromechanics. Using state-of-the-art electromechanical human biventricular simulations, we aim to identify biomarkers that are most representative to the pro-arrhythmic substrate for each state, thus facilitating post-MI risk stratifications to go beyond LVEF. We hypothesise that different ECG abnormalities are explained by different degrees of ionic current remodelling leading to activation sequence abnormalities, dispersion of repolarisation, early after depolarisations (EADs) and alternans, whereas LVEF is insensitive to ionic current remodelling that underpins ECG disease markers and reflects predominantly calcium transient and structural abnormalities.

## Results

### Human modelling and simulation for ECG phenotypes in acute and chronic post-MI

*Figure 1* demonstrates the ability of human electromechanical simulations to reproduce a variety of clinically reported phenotypes in patients with acute and chronic infarction, in agreement with clinical measurements of ECG and pressure-volume biomarkers, as quantified in further detail in (*Appendix 1—table 7*, details of clinical database in Appendix 1.5). When imposing acute post-MI remodelling, simulated ECGs reproduced fractionated QRS complexes, T-wave inversion, Brugada phenocopy ST-segment elevation and QT interval prolongation in the anterior leads (*Figure 1A*), which are common ECG phenotypes observed in acute post-MI patients. Simulations also recapitulated ECG morphology similar to healthy subjects with upright T-waves (*Figure 1A*, right), which can also be present in acute post-MI.

In chronic post-MI, simulated ECG displayed upright T-waves, and global prolongations of QT intervals in all precordial leads (*Figure 1B*, top row), which are characteristic of chronic patients with worse clinical outcomes (*Ahnve, 1985*; *Oikarinen et al., 1998*). A range of T-wave durations were present, as can be found in chronic post-MI (*Figure 1B*, bottom row). Appendix 1.5 summarises the clinical ECGs used for comparisons in *Figure 1*. The full 12-lead ECGs for acute and chronic post-MI simulations can be found in *Appendix 1—figures 7 and 8*.

Simulations using the ventricular population of models showed that the described ECG features of acute and chronic post-MI were mostly preserved across variations in ionic current conductances (*Figure 1C*). Sensitivity analysis showed first that large changes in apex-to-base and transmural heterogeneities only altered T-wave amplitude but not its polarity and did not affect the ST-segment (*Appendix 1—figures 3 and 4*), and second that changes in mechanical parameters did not affect the ECG morphology (*Appendix 1—figures 5 and 6*). This result supports the specificity of post-MI signatures to underlying ionic current remodelling.

### In acute MI, T-wave inversion and Brugada phenocopy can indicate reversed transmural repolarisation gradient and activation failure

Analysis of simulation results enabled the uncovering of specific contributions of different degrees of ionic current remodelling to ECG phenotypes identified in acute versus chronic MI (*Figure 2* vs *Figure 3*). In acute MI, T-wave inversion phenotype was associated with a reversed transmural repolarisation gradient (*Figure 2A*, repolarisation time map insets), due to a delayed activation time and a 57ms APD prolongation at the epicardial BZ compared with control (*Figure 2B*, membrane potential). APD prolongation originated due to inhibition of multiple potassium currents caused by BZ1 ionic remodellling (*Figure 2C*, the first column of $I_{Kr}$, and *Appendix 1—table 4* Acute BZ1).

Brugada phenocopy ECG phenotype was also observed in acute MI with BZ2 remodelling, causing regions of activation failure in the epicardial border zone (*Figure 2A*, the activation and repolarisation time maps), as well as delayed repolarisation in the BZ near the apex, caused by a delayed activation time and a 29ms of epicardial APD prolongation compared with control (*Figure 2B*, membrane potential). These were caused by strong inhibitions of sodium, calcium, and potassium ionic currents

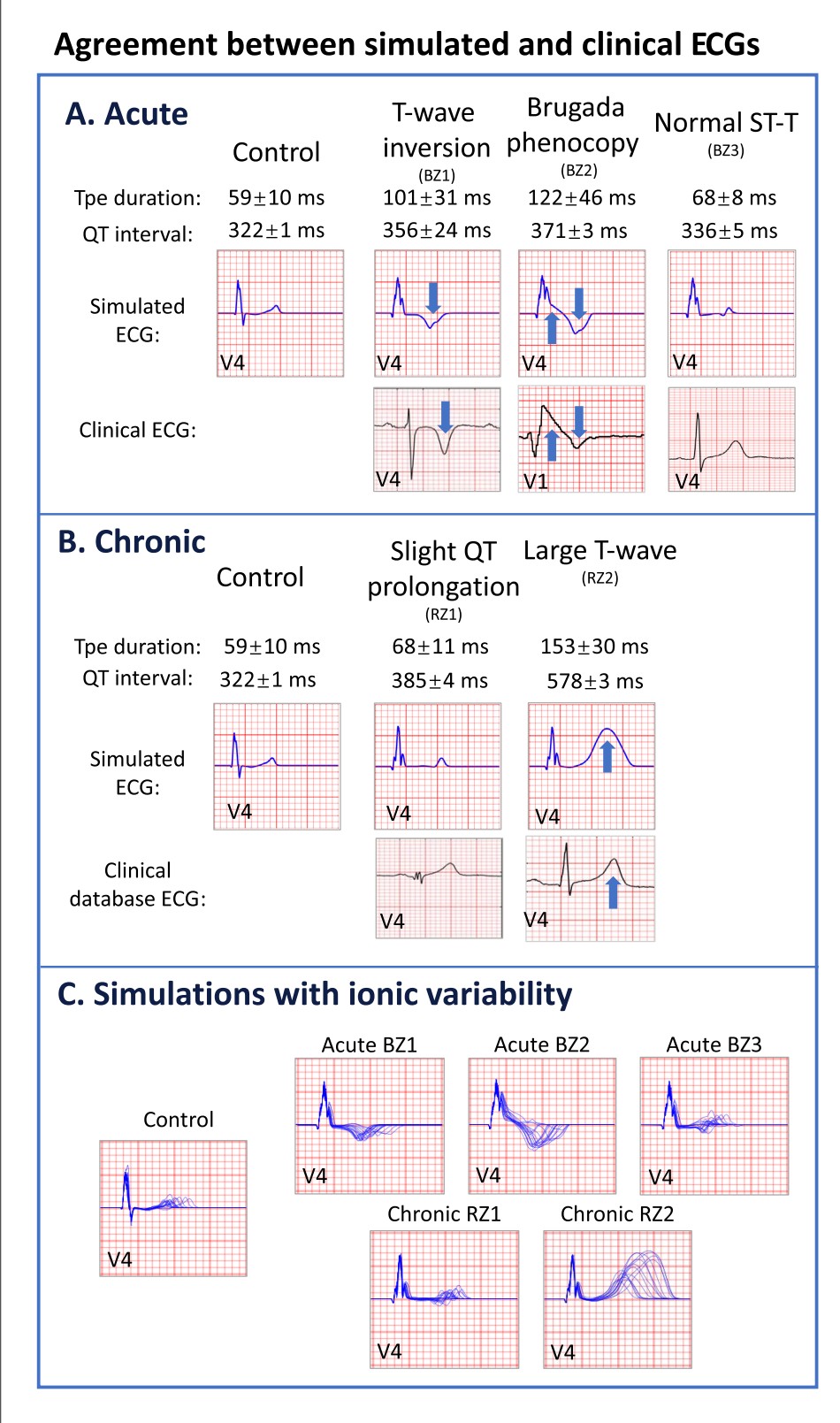

**Figure 1.** Agreement between simulated and clinical ECGs demonstrating variability in clinical phenotypes in acute and chronic post-myocardial infarction (post-MI). (**A**) In acute MI, simulated ECGs show T-wave inversion (border zone model 1 (BZ1)), Brugada phenocopy (BZ2), and normal phenotypes (BZ3), in accordance with phenotypes found in clinical databases. (**B**) In chronic MI, simulated ECGs show prolonged QT and upright

*Figure 1 continued*

T-waves with a range of amplitude and duration (remote zone model 1 and 2 (RZ1, RZ2)) comparable to those observed in clinical databases. (**C**) ECG simulations of control, and acute and chronic post-MI considering ionic current variability of the baseline ToR-ORd model. T wave morphologies for acute and chronic post-MI are mostly preserved across ionic variability.

in the BZ (*Figure 2C*, the second column of $I_{Na}$, $I_{CaL}$, $I_{Kr}$, and *Appendix 1—table 4* Acute BZ2). In this simulation, electrical activation in the infarcted region was preserved despite the conduction block in the BZ because of higher expressions of the L-type calcium channel in the mid-myocardium (see *Appendix 1—figure 11*).

Acute MI with upright T-waves corresponded to a comparable transmural repolarisation gradient as in control with BZ3 ionic current remodelling (*Figure 2A*, the repolarisation time map). In this case, the slight shortening of APD in the epicardial BZ partially compensated for the activation delay in the BZ and therefore resulted in negligible changes in the T-wave of the ECG (*Figure 2C*, the third column of $I_{Kr}$, and *Appendix 1—table 4* Acute BZ3).

### In chronic post-MI, variable T-wave width can be explained by the extent of repolarisation dispersion between border zone and remote zone

In chronic MI, global QT prolongation was due to APD prolongation in the remote myocardium (*Figure 3*, repolarisation time maps and membrane potentials). Recovery of the upright T-wave in the anterior leads (compared to acute MI) was due to a recovery of the transmural repolarisation gradient (*Figure 3A*, repolarisation time maps, and *Appendix 1—figure 10*), given the milder $I_{Kr}$ inhibition in the border zone (*Figure 3B*, the first column of $I_{Kr}$, and *Appendix 1—table 4* Chronic BZ). Furthermore, T-wave duration in this stage was mainly determined by the gradient between remote and border zone repolarisation times (*Figure 3C*, repolarisation time maps), where more severe APD prolongation in the RZ led to larger repolarisation gradients and, consequently, larger T-wave duration and amplitude (*Figure 1B*). Specifically, there was an APD difference of 157ms between remote and border zone cell models for the large T-wave case versus only 12ms for the slight QT-prolongation case, which accounts for the differences in T-wave peak-to-end duration (162ms vs 72ms) and QT intervals (565ms vs 380ms) between these two cases.

### LVEF failed to indicate the extent of post-MI repolarisation dispersions

In our simulations, acute MI ionic current remodelling yielded mildly reduced LVEF in the baseline ventricular models (43%–47%) compared with control (53%) (*Figure 4A*). LVEF reductions were caused by contractile dysfunction (*Figure 4A*, active tension) due to lowered calcium amplitude in BZ (*Figure 2B*, intracellular calcium transient), which was directly caused by inhibitions of $I_{CaL}$ in all acute phenotypes (*Figure 2C*, $I_{CaL}$, and *Appendix 1—table 4*). A complete loss of contractile function in the BZ resulted in a more severe reduction in LVEF in acute post-MI (to 40% for all acute phenotypes in *Appendix 1—figure 9*). Stroke volumes of the left and right ventricles were well-matched in control conditions (1 mL difference, see *Appendix 1—table 9*), and introducing myocardial infarction caused a decrease of stroke volume in the left ventricles in both acute and chronic MI (see *Appendix 1—table 9*).

Activation failure in Brugada phenocopy caused loss of contractility in the affected regions in the BZ (*Figure 2A*, activation and repolarisation time maps) and resulted in a more severely reduced LVEF to 43% with a substantial non-contracting region with significant wall thinning (*Figure 4A*, systolic wall thinning). Acute MI with no observable ECG abnormalities can still have reduced mechanical function, as measured by an LVEF of 47% (*Figure 4A*, the last column). Comparison between this phenotype and the inverted T-wave phenotype shows that the mechanical dysfunction can be dissociated from repolarisation gradient and T wave abnormalities.

Chronic post-MI simulations showed mild reduction in peak systolic pressure (by 1 kPa) and some reduction in LVEF (*Figure 4B*), which were unaffected by differences in repolarisation gradients. This is because there is a consistent reduction in calcium transient amplitude in the remote zone that is independent of the extent of APD prolongation (*Figure 3B*, intracellular calcium transient and membrane

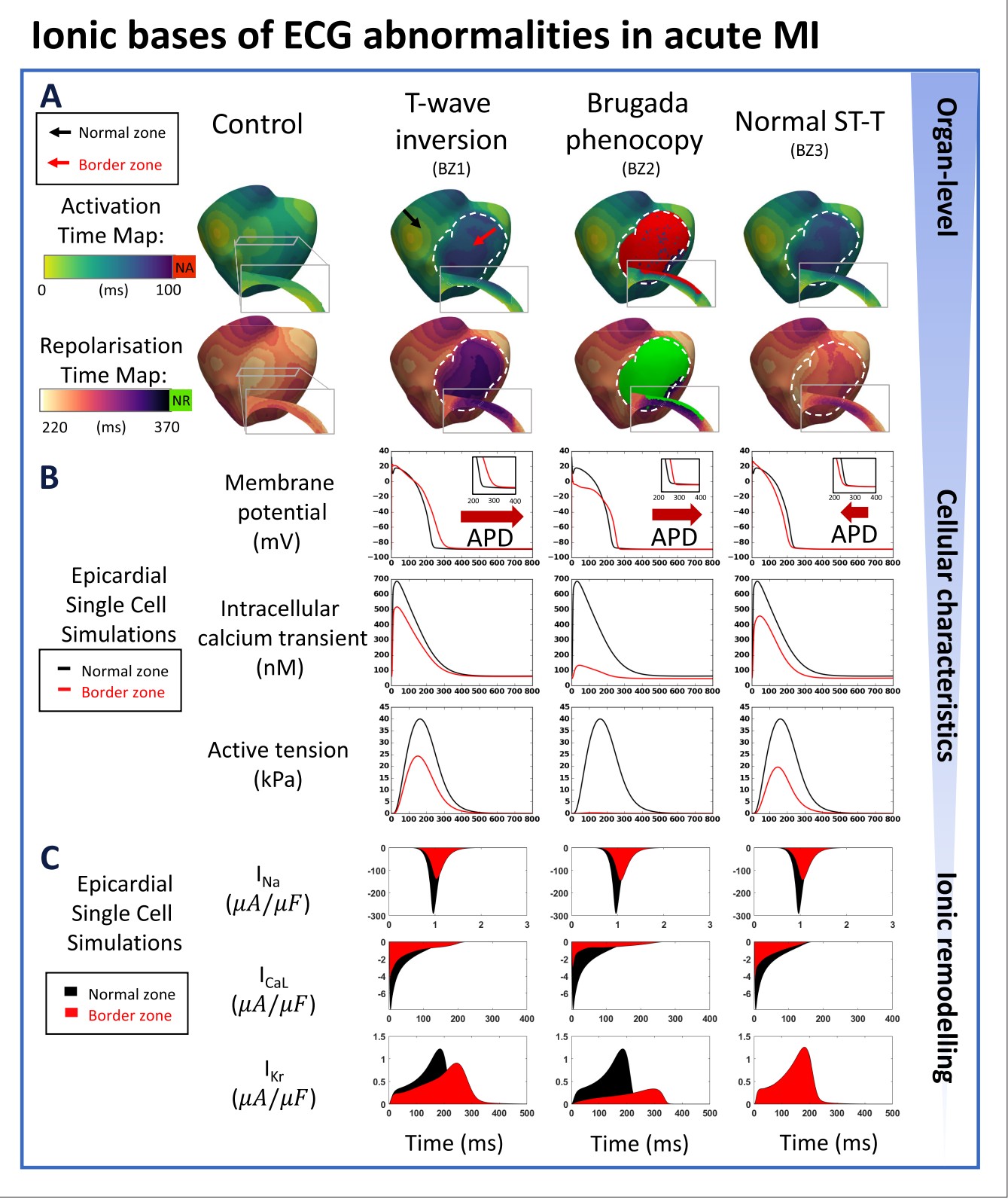

**Figure 2.** Multiscale explanation of ST and T-wave phenotypes in acute MI. (**A**) Activation time maps reveal conduction delay in acute border zone in T-wave inversion and normal ST-T phenotypes, and conduction block in Brugada phenocopy, as well as large repolarisation dispersion and altered transmural repolarisation gradient in T-wave inversion and Brugada phenocopy. Red in activation map show regions of no activation (NA), green in repolarisation map highlights regions of no repolarisation (NR). (**B**) Action potential duration (APD) prolongation is present in T-wave inversion and

*Figure 2 continued on next page*

*Figure 2 continued*

Brugada phenocopy cellular phenotypes, while slight APD shortening is present in normal ST-T (red arrows). Decreased calcium amplitude occurs in all phenotypes, with a corresponding decrease in active tension generation. (**C**) $I_{Na}$, $I_{CaL}$, and $I_{Kr}$ remodelling underpin reduced conduction, reduced calcium amplitude, and alterations in action potential duration, respectively, in all acute phenotypes.

voltage). Therefore, both the acute and chronic electromechanical simulation results showed that the post-MI repolarisation dispersion were not reflected by LVEF.

For both acute and chronic post-MI, simulations done using the population of ventricular models showed similar changes to the PV loop as the baseline model across variabilities in ionic current conductances.

## T-wave alternans and abnormal wave propagation are caused at fast pacing by cellular alternans and EADs, without reduced LVEF at resting heart rate

Increased incidence of T-wave alternans is commonly observed in post-MI patients, and abnormally propagating waves generated from post-MI electrophysiological heterogeneity can trigger lethal arrhythmic events. T-wave alternans were reproduced in the ventricular chronic MI simulations at fast pacing (*Figure 5A*), with RZ2 remodelling, and their mechanisms were revealed through analysis of the high spatio-temporal resolution of simulation data. *Figure 5A* shows upright T-wave morphology and preserved LVEF of 49% at resting rate of 75 bpm (CL = 800ms). However, at fast rates (120 bpm, CL = 500ms), significant beat-to-beat ST and T-wave morphology alterations were observed. This was due to large alternans seen in mid-myocardial single cell simulations of the remote zone at CL of 500ms (*Figure 5C*, green trace), with EAD-driven alternans. These results support the importance of stress tests, since alternans in APD and T-wave can occur at fast heart rates with no sign of LVEF abnormalities at resting heart rate. This is consistent with reports that T-wave alternans under supine bicycle exercise testing was found to be predictive of arrhythmic event after acute post-MI (*Ikeda et al., 2000*).

Simulations with the population of virtual cardiomyocytes models revealed that in addition to the EAD-driven alternans (*Appendix 1—figure 14*), classical calcium-driven alternans were also observed in the population of cell models (*Appendix 1—figures 12 and 13*). The key ionic current remodelling underlying calcium-driven alternans include enhanced CaMKII activity and slower calcium release, as well as suppressed SERCA pump activity in the chronic MI, which are consistent with previous studies (*Livshitz and Rudy, 2007*; *Zhou et al., 2016*; *Tomek et al., 2018*; *Appendix 1—figure 15*). $I_{KCa}$ enhancement in the chronic MI suppressed alternans generation (detailed analysis provided in *Appendix 1—figures 16 and 17*). The median of calcium amplitude was larger in the alternans models than in the non-alternating post-MI models (*Appendix 1—figure 19*), in agreement with the preserved LVEF in the simulations. We did not simulate the effect of this classical calcium-driven alternans on the ECG because the higher pacing rate at which this phenomenon occurs requires the model to include beta-adrenergic inotropic effects to preserve realistic systolic mechanical function.

Another case of chronic MI simulation showed prolonged QT interval of 669ms and preserved LVEF of 49% at 75 bpm (*Figure 6A*). At 120bpm, in this case, ECG reflected chaotic activity (*Figure 6A*, fast pacing simulation) and loss of coordinated mechanical function. Abnormal electrotonic waves were caused by large repolarisation dispersions.

When isolated cells that showed EADs were embedded in the RZ of a ventricular simulation at fast pacing, we saw ectopic wave propagation. This was because the EADs in the RZ generated conduction block, which enabled a large repolarisation gradient to form between the BZ and RZ, thereby leading to ectopy (*Figure 6B*). By the end of the first heartbeat (500ms), the anterior epicardial BZ was fully repolarised (the red trace), but the RZ showed EADs (green, purple and yellow traces). This means that after the second beat stimulus, at 610ms (*Figure 6B*), the long APD of the RZ prevented full activation in the postero-lateral LV (*Figure 6B*, green and purple marker) while the BZ APD was shorter than RZ (*Figure 6B*, red marker) and so could be fully activated. This led to a significant membrane potential gradient between the BZ (red marker) and RZ (green marker), leading to ectopic wave generation due to injury current caused by electronic effects at 710ms in the boundary between BZ and RZ (*Figure 6C*, compare red and green traces) (*Dutta et al., 2016*). This ectopic wave propagated in an anti-clockwise fashion when viewed from the base from 810 to 910ms (*Figure 6B*, basal

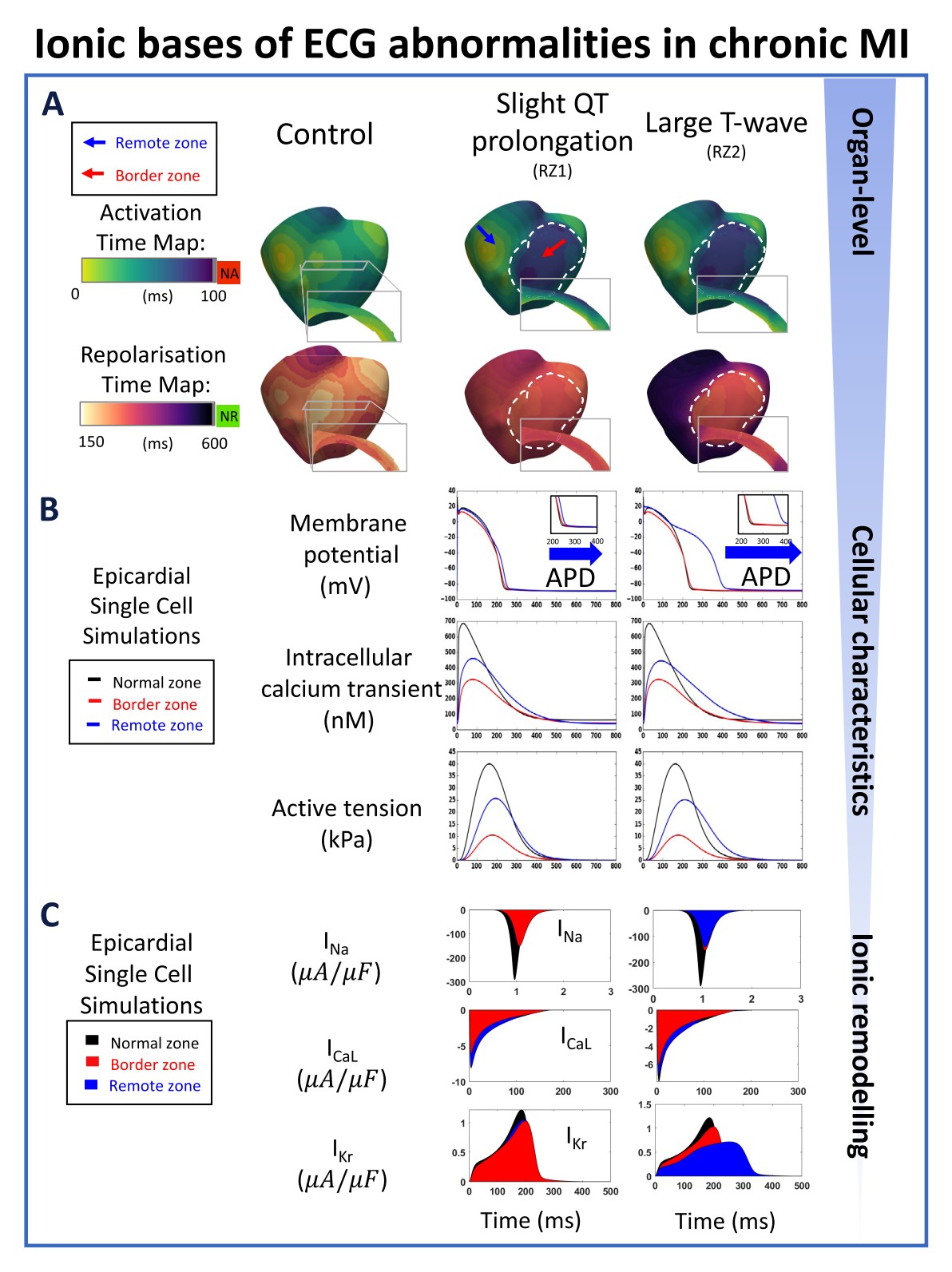

**Figure 3.** Multiscale explanation of QT and T-wave phenotypes in chronic MI. (**A**) Conduction delay in chronic border zone occurs in slight QT prolongation and large T-wave phenotypes, while large repolarisation dispersion exists only in large T-wave. Red in activation map show regions of no activation (NA), green in repolarisation map show regions of no repolarisation (NR). (**B**) Varying degrees of action potential duration (ADP) prolongation in the remote zone (RZ) corresponding to extent of QT prolongation (blue arrows), with decreased calcium amplitude in remote and border zone of both

*Figure 3 continued on next page*

*Figure 3 continued*

phenotypes, and corresponding decrease in active tension generation. (**C**) As in acute MI, $I_{Na}$, $I_{CaL}$, and $I_{Kr}$ remodelling underpin reduced conduction, reduced calcium amplitude, and degree of prolongation in action potential duration, respectively, in both chronic phenotypes.

view). In the right ventricle at 610ms, incomplete repolarisation caused a smaller action potential to be elicited by the sinus stimulus (*Figure 6C*, yellow trace), thus allowing successful propagation of the ectopic wave from the left to the right ventricle from 910 ms to 1010 ms (*Figure 6B*).

This episode illustrated that EADs present in the epicardial remote zone cell model (*Figure 6D*) resulted in large APD dispersion between BZ and RZ, which functioned as the trigger of ectopic wave propagation due to electrotonic gradients. Therefore, the prolonged global QT interval with large T-wave duration and amplitude in leads facing the infarct can be indicative of the risk of large repolarisation dispersion, while the LVEF can be preserved at rest.

Spontaneous EADs were frequently observed in the chronic MI cellular population of models (*Figure 6D* population of models, *Appendix 1—figure 20*), due to less $I_{CaL}$ inhibition in the chronic MI (left, *Appendix 1—table 4*). The key underlying combination of ionic current remodelling for EAD generation include the inhibition of $I_{Kr}$ and the enhancement of $I_{NaL}$, which facilitate reactivation of $I_{CaL}$ (*Appendix 1—figure 21*). Additional contribution of baseline $I_{CaL}$, $I_{Kr}$, and $I_{NCX}$ conductances are consistently observed in all chronic EAD populations (*Appendix 1—table 11*). Similarly, the cellular models that showed EADs also had larger calcium amplitude (*Appendix 1—figure 22*), suggesting a preserved LVEF.

## Discussion

In this study, human electromechanical modelling and simulation enables quantification of the contribution of electrophysiological abnormalities to clinical phenotypes in post-MI patients, from ionic to whole-organ dynamics (summarised in *Table 1*). The credibility of the human electromechanical models and simulation results is supported by their consistency with experimental and clinical data from ionic dynamics to ECG and LVEF biomarkers in healthy, acute and chronic post-MI conditions. Diverse clinical ECG phenotypes are reproduced in the simulations with different degrees of experimentally reported ionic current remodelling for acute and chronic MI; their signature on the LVEF is however weak, with only a small reduction observed. The simulated clinical ECG and LVEF phenotypes were found to be consistent across physiological variabilities in ionic current conductances in the baseline electrophysiological model. Key findings include:

1. In acute MI, T-wave inversion, Brugada phenocopy, and QT prolongation were explained by reversed transmural dispersion of repolarisation in the border zone and infarct, conduction abnormality in the border zone, and increased repolarisation time in the border zone, respectively. (2) In chronic MI, large T-wave duration and amplitude reflects large repolarisation dispersion between remote and border zones, and global QT prolongation was caused by AP prolongation in the remote zone.
2. Reduction in LVEF is ubiquitous across acute and chronic post-MI phenotypes and can be explained by decreased intracellular calcium amplitude and activation failure (in the case of Brugada phenocopy). This effect was independent of changes in the dispersion of repolarisation.
3. Interestingly, fast pacing simulations with T-wave alternans or abnormal propagation driven by cellular alternans or EADs both showed preserved LVEF at resting heart rate, highlighting the fact that preserved LVEF at rest does not guarantee low arrhythmic risk.

Collectively, our results show the proarrhythmic post-MI electrophysiological dispersions caused by cellular remodelling of ionic currents are reflected in QT and T wave morphology biomarkers rather than in LVEF, which questions the use of LVEF as the dominant biomarker in clinical risk stratifications.

### Acute MI T-wave inversion and Brugada phenocopy are caused by reversed transmural repolarisation gradient and regional conduction abnormality

Three distinct types of T-wave morphology were generated by our acute MI biventricular simulations: T-wave inversion, Brugada-phenocopy and normal upright T-wave. We obtained them by applying three types of acute BZs in ventricular simulations, considering both APD prolongation and

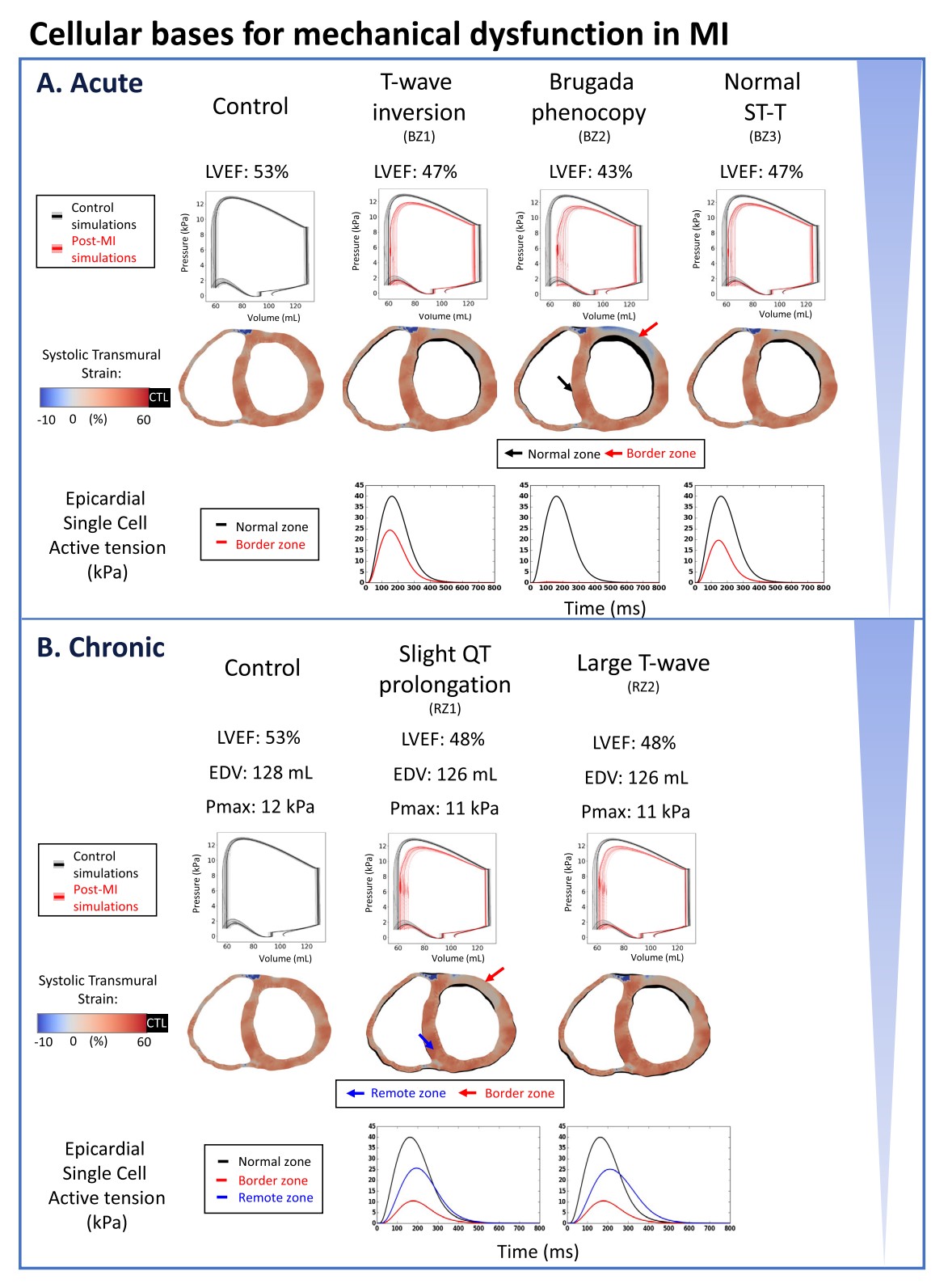

**Figure 4.** Reduced LVEF and heterogeneous systolic deformation caused by ionic current remodelling in both acute and chronic post-myocardial infarction. Pressure-volume loops are shown in black (control) or red (post-MI) traces for the baseline model, and in gray (control) or pink (post-MI) traces for the population of models. (**A**) Reduced LVEF in all acute phenotypes due to reduced active tension amplitude in the border zone (BZ1~3). Brugada phenocopy shows the lowest LVEF due to activation block and loss of contractile function in part of the border zone in addition to reduced

*Figure 4 continued on next page*

*Figure 4 continued*

active tension amplitude in the activated border zone due to ionic current remodelling. Reduced contractile function in infarct and border zone results in infarct thinning and bulging in systole. Systolic cross section of control simulation shown in black (CTL) with post-MI cross-sections superimposed. (**B**) Reduced LVEF in both chronic phenotypes due to reduced active tension amplitude in remote zone (RZ) and border zones, independent of the extent of QT prolongation (RZ1, RZ2). Scar stiffening helped to reduce infarct bulging. Systolic cross-section of control simulation shown in black (CTL) with post-MI cross-sections superimposed.

shortening, as a reflection of the variable experimental results (*Mendonca Costa et al., 2018*). Collectively, our results highlighted the importance of investigating the implications of the various degrees of experimentally-reported ionic current remodelling to explain phenotypic variability of patients with MI. T-wave inversion is a commonly observed feature in the acute MI patients, and is commonly associated with arrhythmic risk (*Tikkanen et al., 2015*). Here, we showed the reversed transmural repolarisation gradient caused by delayed activation and APD prolongation in the epicardial BZ accounted

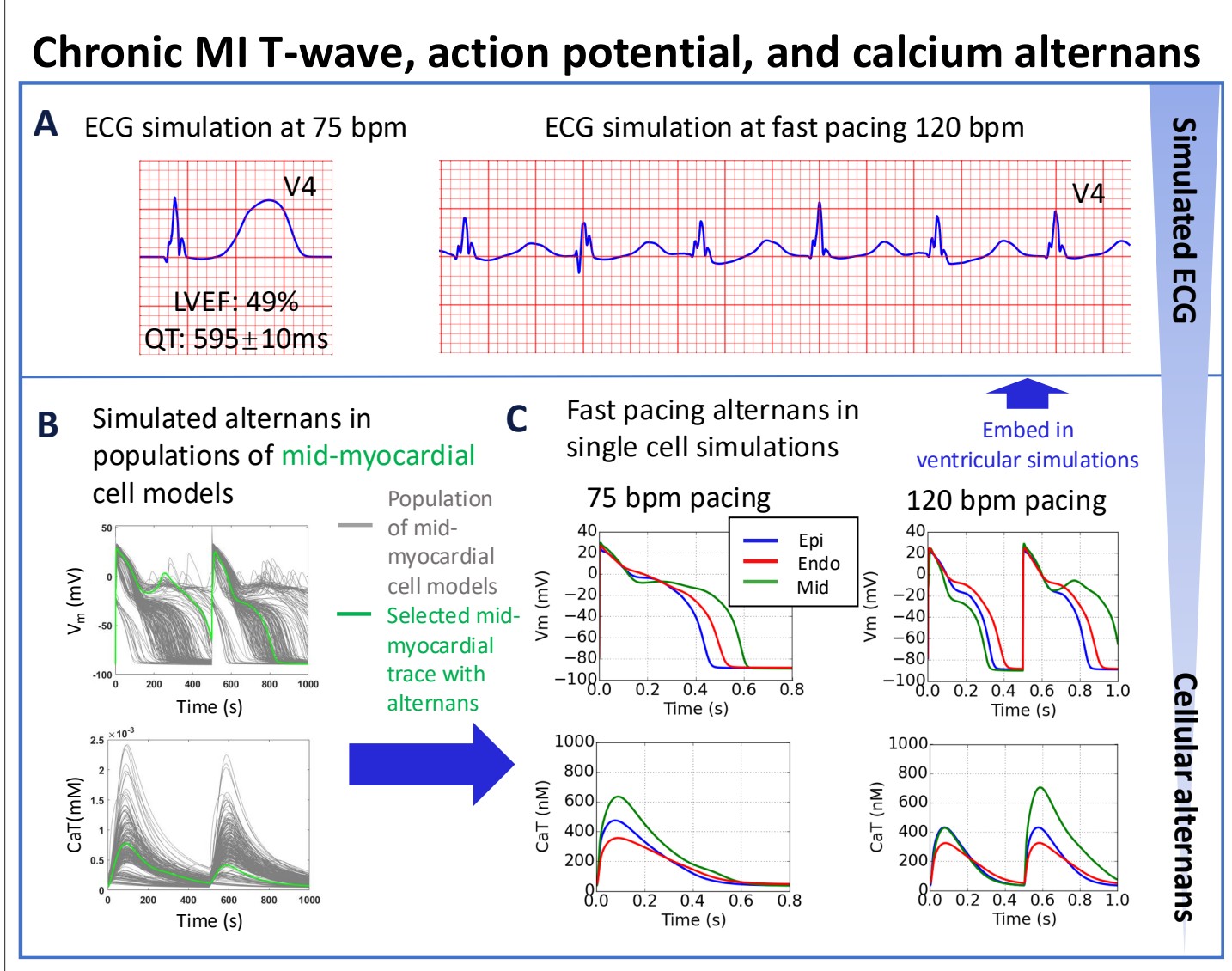

**Figure 5.** T-wave alternans in simulations underpinned by APD and calcium alternans at fast pacing (120 bpm), albeit with preserved left ventricular ejection fraction (LVEF = 49%) at rest (75 bpm) for the chronic MI phenotype (**A**). (**B**) Simulated APD and calcium traces in midmyocardial population of models with remote zone 2 (RZ2) remodelling. (**C**) Large action potential and calcium transient alternans were caused by EADs in simulations at 120 bpm with midmyocardial cells affected by RZ2 ionic current remodelling (green traces, representative example at 75 vs 120 bpm). A single cell model (in green) was selected from the population of models (in grey) for embedding into the remote region for ventricular simulations.

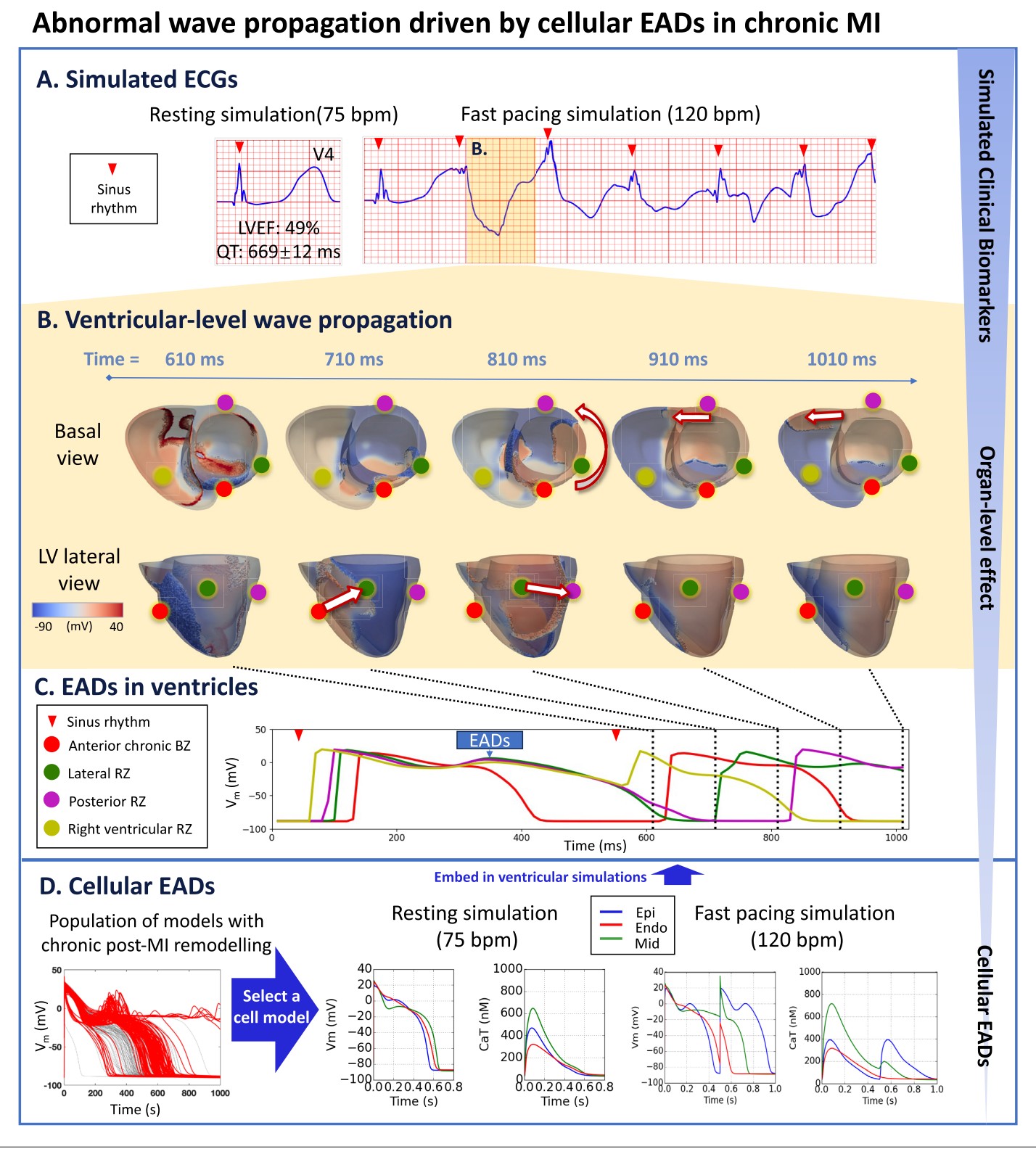

# Abnormal wave propagation driven by cellular EADs in chronic MI

**Figure 6.** Prolonged QT and preserved LVEF at rest can manifest as severely abnormal ECG at fast heart rates in chronic MI with RZ2. (**A**), due to electrotonically-triggered EADs across the border zone (**B**). In (**B**), membrane potential changes for the first 1010ms of fast pacing simulation, showing ectopic wave generation driven by electrotonic gradient at 710ms (arrow from red dot to green dot in lateral view), and anticlockwise propagation of ectopic wave starting at 810ms (anticlockwise arrow in basal view, and arrow from green dot to purple dot in lateral view). Ectopic wave propagates

*Figure 6 continued on next page*

*Figure 6 continued*

towards the right ventricle via the posterior side at 910ms (arrow in basal view) and at 1010ms (arrow in basal view). (**C**) Local action potential at anterior (red), lateral (green), posterior (purple), and right ventricular (yellow) sites. (**D**) A population of models demonstrating chronic remote zone 2 (RZ2) remodelling in promoting EADs. A representative example was selected from the population of models that showed EAD and was embedded in ventricular simulations.

for this phenotype. The link between transmural repolarisation gradient and T-wave polarity has been reported previously (*Okada et al., 2011*) and is consistent with our results. Brugada-phenocopy was also observed in some acute MI patients (*Anselm et al., 2014*), and our simulation results showed it could be a reflection of regional conduction abnormality combined with APD prolongation. Although some animal experiments showed acute MI BZ APD shortening (*Mendonca Costa et al., 2018*), we showed the prolongation of BZ APD was underlying the QT prolongation, T-wave inversion and Brugada phenocopy in the leads facing the infarct (*Appendix 1—figure 7* showing lead dispersions), which is consistent with the QTc prolongation observed in the anterior leads of acute anterior infarction patients (*Guaricci et al., 2018*). Apart from the above, normal T-waves and QT intervals were also commonly observed in patients post percutaneous coronary intervention, which can be reproduced when the post-MI repolarisation dispersion was small between BZ and NZ (BZ3). It is worth noting that, in addition to having a mild border zone remodelling as shown in BZ3, a silent ECG signature can also be due to a reduced transmural extent of the infarct, as has been shown in previous computational studies (*Loewe et al., 2018*; *Wang et al., 2021*). Therefore, T-wave inversion, Brugada phenocopy and QT prolongation occur in the leads facing the infarct can be useful biomarkers indicating bigger repolarisation dispersions and/or larger transmural extent in the acute MI.

## Wide and tall T-wave is explained by large repolarisation dispersions between BZ and RZ in healed post-MI hearts

Our simulated chronic ECGs recapitulated the recovery of T-wave polarity observed in patients after a period of healing. This was achieved through the recovery of the transmural repolarisation gradient caused by the milder $I_{Kr}$ inhibition in the chronic BZ (*Hegyi et al., 2018*). Experimental studies in different species showed inconsistent results regarding the chronic BZ APD (*Mendonca Costa et al., 2018*). Our chronic BZ remodelling produced slightly longer APD than the NZ, which is consistent with observations in healed human BZ (*Dangman et al., 1982*). However, in minipigs, these remodelling caused shorter BZ APD than in NZ (*Hegyi et al., 2018*). This interesting discrepancy between minipigs and human may be due to the different balance of ionic currents across species, which showed the benefits of human electrophysiology models in overcoming the inter-species differences.

The two types of T-wave morphologies in the simulated chronic ECGs corresponded to different extents of RZ APD prolongations, which were commonly observed in healed RZ of post-MI animals

**Table 1.** Linking clinical ECG and left ventricular ejection fraction (LVEF) phenotypes to tissue heterogeneities and subcellular ionic current remodelling in acute and chronic post-myocardial infarction.

| Clinical Phenotypes | Tissue or Cell Level phenomena | Corresponding Post Infarction Ionic Current Remodelling |
|---|---|---|
| Acute MI T-wave inversion in ECG | Reversed transmural repolarisation gradient due to delayed activation and repolarisation in the epicardial border zone | Inhibition of potassium currents in the border zone as well as the slower transmural conduction velocity |
| Acute MI Brugada phenocopy in ECG | Delayed repolarisation, as well as a small region of activation failure in the epicardial border zone | Strong inhibitions of sodium, calcium and potassium ionic currents in the border zone |
| Chronic MI upright tall T-waves in ECG | Large repolarisation time gradient between remote and border zones caused by more severe delay of repolarisation in the remote zone | More severe potassium channel suppression in the remote zone |
| Chronic MI T-wave alternans | Cellular repolarisation alternans or early afterdepolarisation | Suppressed SERCA and augmented CaMKII activity for alternans; Enhanced late sodium current and suppressed hERG current for early afterdepolarisation |
| Acute MI reduction in LVEF | Reduced calcium amplitude and/or regional conduction block | Inhibitions of calcium and sodium currents |
| Chronic MI reduction in LVEF | Reduced calcium amplitude | Decreased SERCA activity |

(*Hegyi et al., 2018*), and in failing human myocytes (*Li et al., 2004*). The substantial RZ APD prolongation was reflected as global QT prolongation in all leads, and the large APD dispersion between chronic BZ and RZ generated wide and tall T-waves in the precordial leads facing the infarct (*Appendix 1—figure 8* for global and dispersed ECG characteristics). Previous simulation studies also found the T-wave amplitude and area were proportional to the dispersion of repolarisation (*Arteyeva and Azarov, 2017*). Therefore, these results demonstrated that in patients with global QT prolongation, leads with bigger T-wave amplitudes could reflect increased local heterogeneity in repolarisation.

## T wave alternans and severely abnormal ECGs at fast pacing are caused by alternans and EADs in chronic infarction

Post-MI ionic current remodelling promoted alternans generation, which resulted in T-wave alternans in simulated ECGs, consistent with the higher incidence of T-wave alternans reported in post-MI patients (*Martin et al., 2009*). Two types of repolarisation abnormalities were observed in our post-MI models: EADs and alternans. One crucial mechanism promoting alternans behaviour at the cellular level is the increased activity of CaMKII, observed in the acute MI BZ (*Hund et al., 2008*), as well as in the hypertrophied and failing myocardium (*Anderson et al., 2011*). Enhanced CaMKII phosphorylation may preserve the contractility of the heart through the phosphorylation of phospholamban and the L-type calcium channels, but increased RyR phosphorylation by CaMKII resulted in prolonged RyR opening as well as the enhancement of spontaneous calcium sparks, which can contribute to alternans and triggered arrhythmias (*Maier and Bers, 2007*).

Generation of EADs due to chronic post-MI ionic current remodelling, such as $I_{Kr}$ inhibition and $I_{NaL}$ enhancement, was consistent with previous studies (*Coppini et al., 2013*). We also found that both repolarisation reserve remodelling ($I_{NaL}$ and $I_{Kr}$) and calcium system remodelling ($J_{up}$ and CaMKII) are important for the EAD-driven alternans (details provided in *Appendix 1—figures 23 and 24*). EADs in the RZ can create large repolarisation dispersion in the ventricle, facilitating abnormal electrotonic wave propagations. Similar re-entrant waves caused by electrotonic gradients were also observed in previous studies of acute ischaemia (*Ridley et al., 1992*; *Dutta et al., 2016*; *Boukens et al., 2021*).

## LVEF should be combined with QT and T-wave characteristics for arrhythmic risk stratification

In this study, we observe non-structurally induced reductions of LVEF in the both the acute and the chronic post-MI stages. At both stages, ventricles with different extents of repolarisation dispersion may have similar LVEF because they have similar degrees of calcium reduction (acute MI T-wave inversion vs normal ST-T, and chronic MI two cases). Models with inducibility of T-wave alternans and arrhythmia at fast pacing rates may present with a preserved LVEF at resting heart rates (*Figures 5 and 6*). Our cellular level results also showed models with inducibility of repolarisation abnormalities, such as alternans and EADs, tended to have more preserved CaT magnitudes at rest rates. Therefore, these phenomena all support the fact that preserved LVEF measured at rest does not guarantee low arrhythmic risk.

A recent clinical study of post-MI patients with preserved LVEF showed defibrillators are needed in those patients with electrophysiological risk factors, such as prolonged QTc, increased T-wave alternans, to prevent sudden cardiac death (*Gatzoulis et al., 2019*). Consistently, we also found post-MI alternans and EADs can present as alternans of T-wave morphology and prolonged QT intervals. In addition to the global ECG changes, our simulation results also showed increased local repolarisation dispersion can be reflected in the leads facing the infarct: inverted T-wave and prolonged QT in the acute MI, and wide and tall T-wave in the chronic MI. Therefore, we suggest the consideration of these signs as markers of high arrhythmic risk.

### Limitations

The main goal of this study is to investigate phenotypic variability in ECG and LVEF biomarkers arising from post-MI ionic current remodelling. We have shown that the relationship between phenotypic variabilities and ionic current remodelling remains consistent across physiological ranges of variation of the ionic current conductances in the baseline cell model. Other sources of variability were not considered in this study including: heart anatomy, location and timing of the early activates sites, calcium sensitivity, conduction velocity. These could all modulate quantitatively the findings but we

do not anticipate strong implications in the findings. The effect of variability in location and size of the scar on the ECG has been explored elsewhere (*Li et al., 2024*). LVEF reduction in clinical cases are more significant than in our simulations due to factors other than ionic current remodelling: RZ structural remodelling and elevated myocardium stiffness, and abnormalities in anatomy. However, these effects have been well-documented elsewhere and this study serves to elucidate the non-structural mechanisms that underpin LVEF reduction that is linked to electrophysiological remodelling and arrhythmic risk. The basal plane in our simulation was fixed in space, which was necessary due to the segmented geometry from clinical MRI and to prevent unphysiological motion at the truncated basal plane. Despite this limitation, our conclusions regarding the relative comparisons of mechanical dysfunction are likely to still hold.

In addition, post-MI ionic current remodelling can be modulated by other acute and chronic factors such as autonomic modulation (beta-adrenergic effects), inflammation, cell death, and metabolic remodelling, which can be explored in future work. The limitations in this study call for a need for personalised digital twins to be generated in the future to facilitate a better understanding of the interaction between structural remodelling and electrophysiological alterations.

## Conclusions

Human-based electromechanical simulations reveal ionic mechanisms underlying T-wave inversion, Brugada phenocopy, and upright T-wave in acute post-MI, as well as the upright T-wave, QT prolongation and T-wave alternans in the healed chronic MI. In acute MI, while the potassium current reduction in the border zone was implicated for all ECG abnormalities, the more severe ECG abnormalities in the Brugada phenocopy implicates additional remodelling for the sodium and calcium currents, which were also key factors in reduced mechanical function. In chronic MI, the degree of QT prolongation and the generation of pro-arrhythmic injury currents were directly related to the severity of potassium current remodelling in the remote region. In addition, late sodium current remodelling could be an important factor underpinning T-wave alternans in chronic MI through the promotion of EAD-driven alternans. Our results show that T-wave inversion, wide and tall T-wave, and QT prolongation in the leads facing the infarct are indicative of local dispersion of repolarisation, which is independent from the reduction of LVEF. Our simulation results suggest the utilisation of T-wave morphology, T-wave alternans and QT prolongation to improve risk stratification biomarkers even when the resting LVEF is preserved.

## Materials and methods
### Human multi-scale ventricular electromechanical modelling and simulation: from ionic current remodelling to ECG and LVEF

A human ventricular electromechanical modelling and simulation framework is constructed using a population of models approach and evaluated using experimental and clinical data to enable the investigations of variable post-MI patient phenotypes, from ionic current remodelling to body surface ECGs and pressure-volume (PV) loops (*Figure 7A*). A cardiac magnetic resonance (CMR)-based biventricular anatomical mesh (*Figure 7B*) with corresponding torso geometry was used for all simulations in this study, with an anterior scar that is 75% transmural (*Wang et al., 2021*). Electrical propagation was simulated using the monodomain equation with orthotropic diffusion based on rule-based fields for fibre directions (*Streeter et al., 1970*) with sheet directions normal to the endocardial/epicardial surface (*Levrero-Florencio et al., 2020*; *Figure 7B*). Transmural and apex-to-base heterogeneities (*Mincholé et al., 2019*) were introduced and a sensitivity analysis was performed to investigate their implications in ECG biomarkers and LVEF (*Figure 7B*). Electrical stimulus via Purkinje-myocardial junctions was simulated by an endocardial fast-activation layer with root node locations to achieve realistic QRS complex morphologies simulated at clinically standard lead locations (*Figure 7C*; *Mincholé et al., 2019*). In healthy tissue (normal zone [NZ]), monodomain diffusivities were calibrated to achieve experimentally measured orthotropic conduction velocities of 67 cm/s, 30 cm/s, and 17 cm/s (*Caldwell et al., 2009*).

Since anterior infarction is very common and related to the worst prognosis (*Stone et al., 1988*), 75% transmural extent of infarct from the endocardial surface was introduced to the anterior myocardial wall (*Figure 7C*) to match the clinical definition of transmural infarctions (*Sato et al., 2008*), with

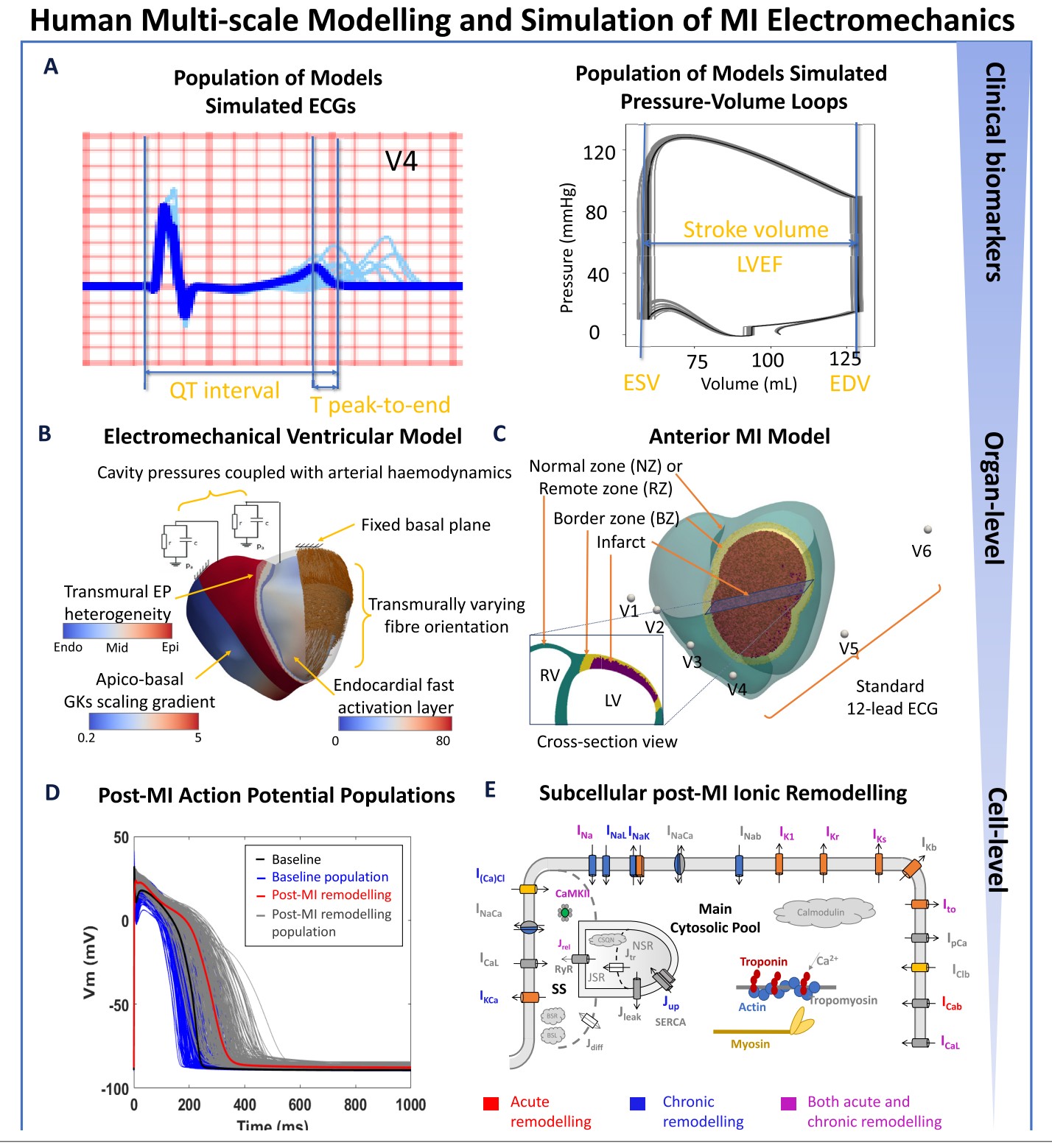

**Figure 7.** Human-based multi-scale modelling and simulation in acute and chronic myocardial infarction. (**A**) Simulations using a population of ventricular models (n=17) to produce ECGs (light blue traces) and pressure-volume (PV) loops (grey traces) superimposed with the baseline ventricular model (ECG in blue and PV in black). Biomarkers are calculated from the baseline simulation of ECG and PV, as illustrated. (**B**) Ventricular electrophysiology is simulated using a fast endocardial activation layer to approximate Purkinje-myocardial junction, experimentally-informed transmural and apico-basal heterogeneities in action potential duration, and transmurally varying myocyte orientation. Mechanical pumping behaviour is modelled

*Figure 7 continued on next page*

*Figure 7 continued*

by coupling the intraventricular pressures with a two-element Windkessel model of arterial haemodynamics with a fixed basal plane. (**C**) An anterior 75% transmural infarction is modelled with acute and chronic ionic current remodelling embedded in the border zone and remote zones. Standard 12-lead ECG was evaluated at standard body-surface locations. (**D**) Simulated action potentials using populations of human ventricular models in healthy (baseline) and acute and chronic post-MI conditions with different degrees of ionic current remodelling. (**E**) Schematic representation of ionic fluxes, calcium dynamics and actin/myosin contraction mechanisms in the human ventricular electromechanically-coupled cellular model.

scar and border zones (maximum width 0.5 cm) taking up 12.4% and 11.7% of ventricular volume (or 15.8% and 15.0% of left ventricular volume), respectively. Maximum border zone width was based on various reports of systolic strain dysfunction in <1 cm proximity to the infarct (*Gallagher et al., 1986*; *Van Leuven et al., 1994*). Electrophysiological remodelling was implemented in the border zone (BZ) and the chronic remote zone (RZ), which covered the entire non-infarcted and non-BZ region in both ventricles. In BZ and infarct zones, the diffusivities were calibrated to reproduce conduction slowing (one-third of the NZ conduction velocities). For each virtual cardiomyocyte, the electromechanical single-cell model considered the human-based ToR-ORd electrophysiological model (*Tomek et al., 2019*; extensively validated in control, disease and drug block conditions) coupled with human excitation-contraction and active tension Land model (*Land et al., 2017*; *Levrero-Florencio et al., 2020*; *Figure 7E*).

The human biventricular model incorporated strongly-coupled electromechanics with orthotropic passive mechanical behaviour and balance of linear momentum with inertial effects, as in our previous work (*Levrero-Florencio et al., 2020*; *Margara et al., 2021*; *Wang et al., 2021*). In brief, firstly, intracellular calcium concentration drives crossbridge cycling and force production through unblocking the crossbridge binding-site. Secondly, calcium sensitivity and force production are a function of fibre stretch ratio. Thirdly, stretch rate affects distortion-dependent cross-bridge unbinding (*Land et al., 2017*) and the diffusivity tensor is transformed using the deformation gradient tensor such that the prescribed values describe conductivities in the deformed state (*Levrero-Florencio et al., 2020*).

An elastic spring boundary condition was set to act perpendicularly to the epicardial surface, to simulate pericardial constraint, and the basal plane was fixed in space to prevent unphysiological tilting and expansion due to the unavailability of closed basal geometry. Pressure boundary condition on the left and right endocardial surfaces were controlled using a series of five equations (see Appendix 1.2) that controls (1) active diastolic inflation followed by electrical activation and (2) isovolumic contraction, (3) ejection coupled with two-element Windkessel aortic haemodynamics model, (4) isovolumic relaxation, and (5) passive inflation and relaxation (*Wang et al., 2021*).

The sheet active tension was set to 30% of fibre active tension to achieve sufficient LVEF in control conditions, based on the results from a sensitivity analysis in Appendix 1.6. In brief, we evaluated the sensitivity of ECG morphology and LVEF to changes in the calcium sensitivity of troponin binding in excitation-contraction coupling and the active tension in the sheet direction as a percentage of that in the fibre direction.

The passive stiffness parameters were calibrated based on a previous sensitivity analysis (*Wang et al., 2021*) to achieve 53% LVEF and a physiological pressure-volume (PV) loop in a control simulation (see *Appendix 1—table 2* and *Appendix 1—table 3* for a list of calibrated parameters).

The active tension was set to zero in the scar to represent the myocyte damage, and the chronic passive stiffness parameters of the infarct region were increased 10-fold to mimic fibrotic scar formation (*Sun et al., 2009*). For each acute post-MI phenotype, a case of complete loss of contractile function (zero active tension) in the BZ was also simulated to evaluate the contribution of other non-ionic current remodelling-related abnormalities on ejection fraction.

## Experimentally-informed single cell and ventricular populations of human post-MI electromechanical models

To account for the inter-subject electrophysiological variability widely observed in clinical data, the baseline human cellular electromechanical ToR-Land model was extended to populations of healthy cellular models. Then, post-MI ionic current remodelling was applied to generate populations of post-MI virtual cardiomyocytes (*Figure 7D*). In addition to the baseline ToR-ORd model, several representative cellular models were selected from the population and implemented into the biventricular electromechanical simulations.

An initial population of human ventricular cell models was constructed based on the ToR-ORd model by varying the conductances or magnitudes of $I_{Na}$, $I_{NaL}$, $I_{to}$, $I_{CaL}$, $I_{Kr}$, $I_{Ks}$, $I_{K1}$, $I_{NaCa}$, $I_{NaK}$, $J_{rel}$ and $J_{up}$ by up to ±50% using Latin Hypercube Sampling (*Appendix 1—figures 1 and 2*). As illustrated in previous studies, small populations of models with proarrhythmic ionic current remodelling achieved similar predictability as large populations with uniform current variations (*Zhou et al., 2019*). Therefore, we chose to start with a small healthy population (n=500) and then introduce multiple combinations of ionic current remodelling to mimic the large post-MI variability. After calibration with human experimental data (*Appendix 1—table 1*) and discarding those manifesting EADs at 1 Hz, 245 sets of endocardial, midmyocardial and epicardial models were accepted as the healthy population (*Appendix 1—figure 2*). From this population, a total of 17 sets of cell models were randomly selected and uniformly embedded in 17 ventricular models, to generate ventricular population of models that produce a variety of ECGs (*Figure 7A*, light blue traces) and PV loops (*Figure 7B*, grey traces).

Several degrees of post-MI ionic current remodelling were collated from a combination of human and animal experimental data with variability in severity of disease to explore whether such variabilities can explain variability in established clinical ECG phenotypes. These remodellings have been applied to the healthy celllular model population (n=245) to generate BZ and RZ populations for both acute and chronic post-MI. For acute post-MI (within a week post-occlusion), three types of BZ remodelling (Acute BZ1-3) were considered based on previous modelling work and experimental canine data collected within 5 days post-MI. The three models of acute border zone remodelling had in common strong inhibition of $I_{Na}$ (60~62%). The BZ2 model had more severe inhibition of $I_{CaL}$ and $I_{Kr}$ than the BZ1 model, alongside other minor differences. The BZ3 model, while having less severe potassium currents inhibition than BZ1, had additional remodellings in CaMKII dynamics, RyR time constants, and $I_{Cab}$. For chronic post-MI, ionic current remodelling measured from minipigs 5 mo post-MI with heart failure were used to generate Chronic BZ (affecting only the BZ) and Chronic RZ1 (also affecting the remote myocardium). Another type of remodelling, Chronic RZ2, was established based on multiple experimental data from failing human cardiomyocytes. The RZ covers the entire myocardium apart from the infarct and BZ in chronic MI simulations. Furthermore, reduction of sodium current and SERCA, with enhanced CaMKII activity and slower calcium release induced by CaMKII activation were also implemented in Chronic BZ, RZ1 and RZ2, as observed in human failing cardiomyocytes (details in *Appendix 1—table 4*). Compared with RZ1, the RZ2 model had a significantly stronger inhibition of potassium currents and a lower repolarisation reserve, alongside other more minor differences. Both human recordings and animal data were used for model evaluation, given the scarcity of human tissue, summarised in *Appendix 1—table 4*; *Appendix 1—table 5*. Action potential, calcium transient, and active tension characteristics of the post-MI models are summarised in *Appendix 1—table 6*. These post-MI ionic current remodellings were then applied to the ventricular population of models (n=17) to explain ECG and PV phenotypes while considering physiological population variability in baseline ionic conductances. These remodelled cell models are embedded uniformly within each region according to a prescribed transmural heterogeneity of 30% endo, 40% mid-myocardial, and 30% epicardial cell types.

## Simulation protocols and biomarker calculation

Human virtual ventricular myocytes were paced at 1 Hz for 500 beats to detect EAD generation. For alternans generation, single cells were paced at cycle lengths (CLs) of 500ms, 400ms, and 300ms for 500 beats, and a ΔAPD greater than 3ms between the last two beats at steady state was defined as alternans.

Biventricular electromechanical simulations were performed at 800 ms CL (75 beats per minute) with 100ms allowed for active diastolic filling prior to endocardial activation for each beat. Three beats were sufficient to achieve converged ECG and PV characteristics at 800 ms CL (see *Appendix 1—figure 7* for ECGs for all beats).

For chronic post-MI, an additional fast pacing protocol was applied with 500 ms CL (120 beats per minute), with 50ms allowed for active diastolic filling for each beat. From these fast pacing chronic MI simulations, two sets of cell models showing EAD and alternans behaviours, respectively, were selected from the population of models and embedded according to transmural heterogeneity in the remote zone for ventricular simulations, to test whether the arrhythmic behaviours at the cellular level can result in arrhythmic behaviour at ventricular level and manifest in the ECG. For this fast pacing

protocol, six beats were necessary to achieve converged ECG and PV characteristics at 500 ms CL (see *Appendix 1—figure 8* for simulated ECGs of all beats).

Clinical biomarkers were quantified from the simulated ECG (including the QT interval, QRS duration, T-wave duration, T peak to T end duration, T onset to T peak duration, and QT dispersion, see definition and method of evaluation in Appendix 1.3 and biomarker results in *Appendix 1—table 8*), and from the simulated PV loop (including end diastolic and end systolic volumes, peak systolic pressures and LV and RV ejection fractions), as well as wall thickening strain (see *Appendix 1—table 8*; *Appendix 1—table 9* for ECG and PV biomarkers for all simulated beats).

## Simulation software and computational framework

Cellular electrophysiological simulations and Latin Hypercube Sampling were performed using bespoke MATLAB codes. Coupled cellular electromechanics, as well as biventricular electromechanics simulations, were performed using the high-performance numerical software, Alya, for complex coupled multi-physics and multi-scale problems (*Santiago et al., 2018*) on the CSCS (Swiss National Supercomputing Centre) Piz Daint supercomputer multi-core clusters, granted through the PRACE (Partnership for Advanced Computing in Europe) project. The simulation input files and Alya executable required to replicate the simulated results are available upon request for scientific investigations.

## Preprint

This manuscript was first published as a preprint: Xin Zhou, Zhinuo Jenny Wang, Julia Camps, Jakub Tomek, Alfonso Santiago, Adria Quintanas, Mariano Vazquez, Marmar Vaseghi, Blanca Rodriguez (2022). [Clinical phenotypes in acute and chronic infarction explained through human ventricular electromechanical modelling and simulations]. bioRxiv. https://www.biorxiv.org/content/10.1101/2022.02.15.480392v3.

## Acknowledgements

The authors would like to acknowledge Dr Erica Dall'Armellina, Dr Arka Das, Dr Chris Kelly, and Dr Lei Wang, for discussions on the study. This work was funded in whole, or in part, by the Wellcome Trust (214290/Z/18/Z). For the purpose of Open Access, the author has applied a CC BY public copyright licence to any Author Accepted Manuscript version arising from this submission. This work was supported by a Wellcome Trust Fellowship in Basic Biomedical Sciences to BR (214290/Z/18/Z), an Oxford-Bristol Myers Squibb Fellowship to XZ (R39207/CN063), the Sir Henry Wellcome Fellowship to JT (222781/Z/21/Z), the Personalised In-Silico Cardiology (PIC) project, the CompBioMed 1 and 2 Centre of Excellence in Computational Biomedicine (European Commission Horizon 2020 research and innovation programme, grant agreements No. 675451 and No. 823712), an NC3Rs Infrastructure for Impact Award (NC/P001076/1), the TransQST project (Innovative Medicines Initiative 2 Joint Undertaking under grant agreement No 116030, receiving support from the European Union's Horizon 2020 research and innovation programme and EFPIA), and the Oxford BHF Centre of Research Excellence (RE/13/1/30181). Furthermore, we acknowledge PRACE for awarding access to the Piz Daint resources at the Swiss National Supercomputing Centre, Switzerland (PRACE-ICEI grants icp005 and icp013, awarded to Dr Alfonso Bueno-Orovio), and JURECA at the Juelich Supercomputing Centre (PRACE-ICEI grant icp019 awarded to Dr Zhinuo Jenny Wang) which are partially funded from the European Union's Horizon 2020 research and innovation programme through the ICEI project under the grant agreement No. 800858.

## Additional information

### Funding

| Funder | Grant reference number | Author |
|---|---|---|
| Wellcome Trust | 10.35802/214290 | Blanca Rodriguez |
| Bristol Myers Squibb | R39207/CN063 | Xin Zhou |

| Funder | Grant reference number | Author |
|---|---|---|
| University of Oxford | R39207/CN063 | Xin Zhou |
| Wellcome Trust | 10.35802/222781 | Jakub Tomek |
| Horizon 2020 | 10.3030/764738 | Blanca Rodriguez |
| Horizon 2020 | 10.3030/675451 | Mariano Vazquez Blanca Rodriguez |
| Horizon 2020 | 10.3030/823712 | Mariano Vazquez Blanca Rodriguez |
| National Centre for the Replacement, Refinement and Reduction of Animals in Research | NC/P001076/1 | Blanca Rodriguez |
| Horizon 2020 | 10.3030/116030 | Blanca Rodriguez |
| University of Oxford | RE/13/1/30181 | Blanca Rodriguez |

The funders had no role in study design, data collection and interpretation, or the decision to submit the work for publication. For the purpose of Open Access, the authors have applied a CC BY public copyright license to any Author Accepted Manuscript version arising from this submission.

## Author contributions

Xin Zhou, Conceptualization, Formal analysis, Validation, Investigation, Visualization, Methodology, Writing – original draft, Writing – review and editing; Zhinuo Jenny Wang, Conceptualization, Software, Formal analysis, Validation, Investigation, Visualization, Methodology, Writing – original draft, Writing – review and editing; Julia Camps, Software, Visualization, Writing – review and editing; Jakub Tomek, Methodology, Writing – review and editing; Alfonso Santiago, Adria Quintanas, Software, Writing – review and editing; Mariano Vazquez, Resources, Software, Writing – review and editing; Marmar Vaseghi, Data curation, Validation, Investigation, Writing – review and editing; Blanca Rodriguez, Conceptualization, Resources, Supervision, Funding acquisition, Writing – review and editing

## Author ORCIDs

Xin Zhou https://orcid.org/0000-0003-2814-3399
Zhinuo Jenny Wang https://orcid.org/0000-0001-5325-909X
Jakub Tomek https://orcid.org/0000-0002-0157-4386
Blanca Rodriguez http://orcid.org/0000-0001-6361-3339

Reviewer #1 (Public review): https://doi.org/10.7554/eLife.93002.3.sa1
Reviewer #2 (Public review): https://doi.org/10.7554/eLife.93002.3.sa2
Author response https://doi.org/10.7554/eLife.93002.3.sa3

# Additional files

## Supplementary files
- MDAR checklist

## Data availability

The current manuscript is a computational study. Source code for cellular simulations and post-processing scripts for the ventricular simulations are available at GitHub (copy archived at *Wang and Wang, 2024*). The input files and binary for Alya simulations are hosted on Zenodo. To replicate the study access to an installation of the code in the Nord supercomputer can be requested to mariano@elem.bio.

The following dataset was generated:

| Author(s) | Year | Dataset title | Dataset URL | Database and Identifier |
|---|---|---|---|---|
| Wang ZJ, Zhou X, Rodriguez B | 2024 | Clinical phenotypes in acute and chronic infarction explained through human ventricular electromechanical modelling and simulations | https://zenodo.org/records/13993395 | Zenodo, 10.5281/zenodo.13993394 |

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

## Appendix 1

### 1. Implementation of the calcium activated potassium current in the ToR-ORd model, and the generation of population of ToR-ORd-SK models

As the calcium activated potassium current ($I_{KCa}$) were reported to be enhanced in heart failure, a new formulation of the $I_{KCa}$ was added into the ToR-ORd model based on published data (*Chang et al., 2013*) to obtain an updated model named ToR-ORd-SK model. Due to the coupling of $I_{KCa}$ channels and the L-type calcium channels (*Zhang et al., 2018*), the ratio of $I_{KCa}$ channels in the subspace was set to be the same as the L-type calcium channels in the model. The conductance of $I_{KCa}$ (gkca) was chosen to get a similar current density ratio between $I_{KCa}$ and $I_{Kr}$ as observed in minipig myocytes (*Hegyi et al., 2018*). The conductance of the background potassium current was scaled to 90% to adapt to the implementation of $I_{KCa}$. The formulation of this new $I_{KCa}$ current is the following:

gkca = 0.003;

ikcan = 3.5;

kdikca = 6.05e-04

Fraction$I_{KCa}$ss = 0.8;

Fraction$I_{KCa}$i = 1-Fraction$I_{KCa}$ss;

$$I_{KCa}\_ss = gkca \times FractionI_{KCa}ss \times \frac{Ca_{ss}^{ikcan}}{Ca_{ss}^{ikcan}+kdikca^{ikcan}} \times (Vm - EK)$$

$$I_{KCa}\_i = gkca \times FractionI_{KCa}i \times \frac{Ca_{i}^{ikcan}}{Ca_{i}^{ikcan}+kdikca^{ikcan}} \times (Vm - EK)$$

$I_{KCa}$ = $I_{KCa}$\_ss +$I_{KCa}$\_i;

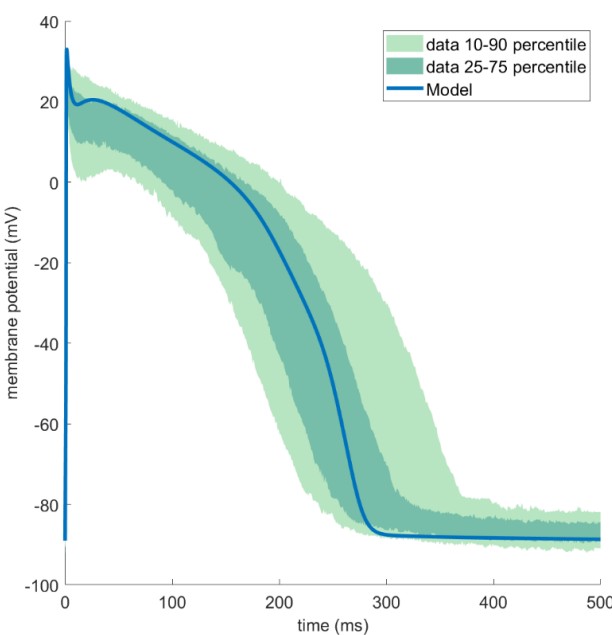

**Appendix 1—figure 1.** The ToR-ORd-SK model produced similar action potential (AP) traces as published human experimental data (*O'Hara et al., 2011*).

An initial population of 500 human endocardial ventricular cell models was constructed based on the ToR-ORd-SK model by varying the conductances or magnitudes of $I_{Na}$, $I_{NaL}$, $I_{to}$, $I_{CaL}$, $I_{Kr}$, $I_{Ks}$, $I_{K1}$, $I_{NaCa}$, $I_{NaK}$, $J_{rel}$, and $J_{up}$ by up to ±50% using Latin Hypercube Sampling, and 253 models were accepted after pacing the models at 1 Hz and calibrated with human experimental data range in *Appendix 1— table 1*. The accepted endocardial parameter scaling factors were applied to the epicardial and midmyocardial baseline models to generate the corresponding epicardial/midmyocardial population of models, and eight models were discarded since they generate early afterdepolarizations (EADs) at 1 Hz in midmyocardial cells.

**Appendix 1—table 1.** Experimental ranges of AP and calcium transient (CaT) biomarkers used to calibrate the normal zone (NZ) endocardial population of models at a pacing cycle length (CL) of 1000ms based on human cardiomyocyte experiments (*Coppini et al., 2013*; *Britton et al., 2017*).

| Biomakers at 1 Hz | minimum | maximum |
|---|---|---|
| Vmax (mV) | 7 | 55 |
| RMP (mV) | –95 | –80 |
| dvdtmax (mV/ms) | 100 | 1000 |
| APD90 (ms) | 180 | 440 |
| APD50 (ms) | | 350 |
| APD40 (ms) | 85 | 320 |
| APD90-APD40 (ms) | 50 | 150 |
| CaTD90 (ms) | 220 | 750 |
| CaTD90 (ms) | 120 | 420 |
| CaTamp (mM) | 2e-4 | 6e-4 |
| CaTmax(mM) | 2e-4 | 10e-4 |
| CaTmin (mM) | 0 | 4e-4 |

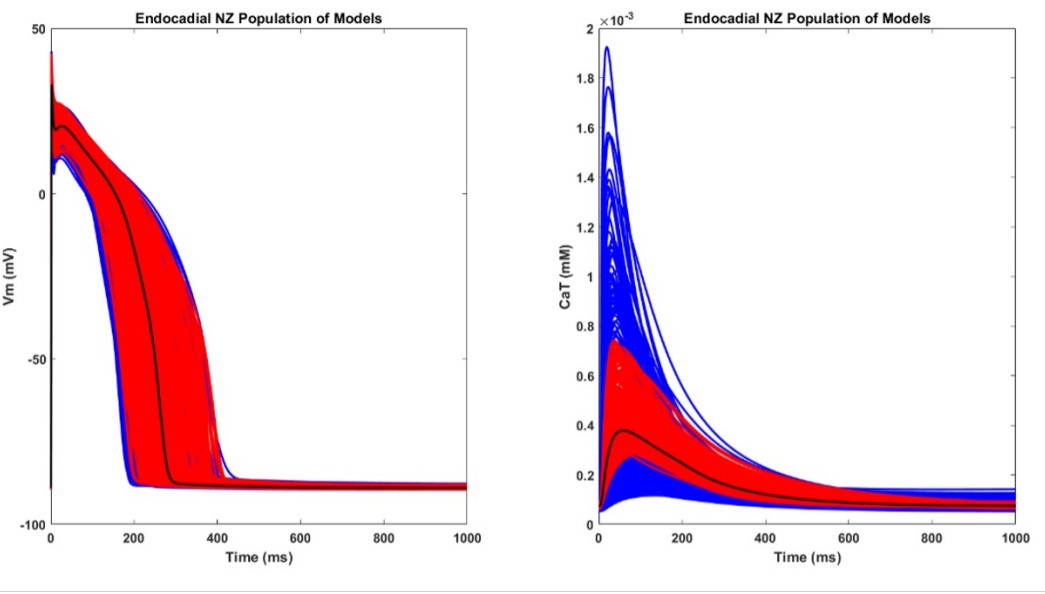

**Appendix 1—figure 2.** The AP and CaT traces of the population of NZ population of models. The blue and red traces are the initial and the accepted population, respectively. The black trace is the baseline endocardial model.

## 2. Human biventricular electromechanical models and calibrated simulation parameters

The parameters to reproduce the healthy baseline and infarcted models are presented in *Appendix 1—table 2*; *Appendix 1—table 3*.

The governing equations are listed here, and described in more detail elsewhere (*Levrero-Florencio et al., 2020*):

$$\chi C_m \frac{\partial V}{\partial t} = \nabla \cdot \left( \boldsymbol{D} \nabla V_m \right) + \chi \left( I_{ion} + I_{stim} \right)$$

$$D = d_f f \otimes f + d_s s \otimes s + d_n n \otimes n$$

where the orthotropic diffusivity tensor $D$ describes different diffusivities along the fibre, sheet, and sheet normal $f, s, n$ directions (values for $d_f, d_s, d_n$ given in **Appendix 1—table 2**), $\chi$ is the surface to volume ratio of the myocardial cell, $C_m$ the membrane capacitance, $V_m$ the transmembrane potential, and $I_{ion}, I_{stim}$ are the ionic channel currents and stimulus current, respectively.

The passive mechanical properties of the myocardium is described in the following strain energy density function $\psi$ in a nearly incompressible form of the Holzapfel Ogden orthotropic model (**Holzapfel and Ogden, 2009**):

$$\psi = \frac{K}{2} (J-1)^2 + \frac{a}{2b} \left( e^{b(I_1 - 3)} - 1 \right) + \sum_{i=f,s}^{2} \frac{a_i}{2b_i} \left( e^{b_i (I_{4i} - 1)^2} - 1 \right) + \frac{a_{fs}}{2b_{fs}} \left( e^{b_{fs} I_{8fs}^2} - 1 \right)$$

where $K$ is the bulk modulus, and the strain invariants $I_1, I_{4f}, I_{4s}, I_{8fs}$ are invariants of the right Cauchy-Green strain tensor $C$, as described below:

$$I_1 = tr C, I_{4f} = f \cdot Cf, I_{4s} = s \cdot Cs, I_{8fs} = \frac{1}{2} (f \cdot Cs + s \cdot Cf)$$

And the values for 'a' and 'b' coefficients are given in **Appendix 1—table 2**.

The active stress $T_a$ produced through calcium-dependent cross-bridge cycling is calculated as:

$$T_a = T_{scale} \cdot h \frac{T_{ref}}{r_s} \left( (\zeta_s + 1) S + \zeta_w W \right)$$

where a scaling factor $T_{scale}$ is applied in the biventricular model to achieve a physiological left ventricular ejection fraction in control, and is given in **Appendix 1—table 2**, $h$ is a function of the fibre stretch ratio that describes the length-dependence of force production, $\zeta_s, \zeta_w$ are state variables describing distortion-decay, $W, S$ are the proportion of cross-bridges in the pre- or post-powerstroke states, respectively, and $r_s$ is a steady-state duty ratio of cross-bridge cycling. Details of this can be found elsewhere (**Land et al., 2017**; **Levrero-Florencio et al., 2020**).

**Appendix 1—table 2.** Calibrated electromechanical parameters for healthy baseline model, and modified parameters for post myocardial infarction models.

| Name | Parameter | Value | Unit |
|---|---|---|---|
| Healthy baseline electromechanical parameters | | | |
| diffusivity in fibre, sheet and sheet normal directions | $d_f$ | 0.00335 | cm/mS |
| | $d_s$ | 0.000723 | cm/mS |
| | $d_n$ | 0.000153 | cm/mS |
| active mechanics: scaling parameter for active tension | $T_{scale}$ | 12 | |
| bulk modulus | K | 12185000 | Ba |
| passive mechanics: exponential term in isotropic matrix, fibre, sheet and normal direction | a | 20000 | Ba |
| | b | 9.242 | |
| | $a_f$ | 30000 | Ba |
| | $b_f$ | 15.972 | |
| | $a_s$ | 20000 | Ba |
| | $b_s$ | 10.446 | |
| | $a_{fs}$ | 10000 | Ba |
| | $b_{fs}$ | 11.602 | |

*Appendix 1—table 2 Continued on next page*

*Appendix 1—table 2 Continued*

| Name | Parameter | Value | Unit |
|---|---|---|---|
| Scar and border zone diffusion parameters | | | |
| | $d_f$ | 0.0012 | cm/mS |
| | $d_s$ | 0.00023 | cm/mS |
| diffusivity in fibre, sheet and sheet normal directions | $d_n$ | 0.000003 | cm/mS |
| Scar mechanical parameters | | | |
| active mechanics: scaling parameter for active tension | $T_{scale}$ | 0 | |
| bulk modulus | K | 12185000 | Ba |
| | a | 200000 | Ba |
| | b | 9.242 | |
| | $a_f$ | 300000 | Ba |
| | $b_f$ | 15.972 | |
| | $a_s$ | 200000 | Ba |
| | $b_s$ | 10.446 | |
| passive mechanics: exponential term in isotropic matrix, fibre, sheet and normal direction | $a_{fs}$ | 100000 | Ba |
| | $b_{fs}$ | 11.602 | |

The pressure boundary condition on the left and right endocardial surfaces are controlled using the following set of piece-wise functions:

(1) *Initialisation*. Both ventricles were firstly inflated to an initial pressure $P_0$ to reach a loaded resting endocardial volume, $V_0$.

(2) *Active inflation*. The pressure in both ventricular chambers is linearly increased to the end diastolic pressure $P_{endd}$ over duration of $t_{diastole}$. This phase mimics the atrial contraction phase of diastolic filling and it is considered the first phase in the cardiac cycle because it follows directly from sinoatrial stimulus.

(3) *Isovolumetric contraction*. Endocardial activation marks the beginning of this phase where stimulated myocytes begin generating active tension and the endocardial pressure (P) is allowed to increase such that the chamber volume (V) is kept approximately constant through the use of penalty terms:

$$dP = -\frac{1}{C_p}dV - \frac{1}{C_v} \cdot \frac{dV}{dt}$$

where $C_p$ and $C_v$ are the penalty terms for volume difference and volume rate, respectively. Furthermore, while $C_v$ is a user defined constant, $C_p$ is defined as: $C_p = \frac{P}{V}$

(4) Ejection. This phase is triggered when the ventricular pressure exceeds the arterial pressure, $P_{art0}$. A two-element Windkessel model is used to model the blood pressure of both the systemic and pulmonary circulation systems during ejection: $C\frac{dP_{art}}{dt} + \frac{P_{art}}{R} = -\frac{dV}{dt}$ where C and R are the compliance and impedance of the circulation systems.

(5) *Isovolumetric relaxation*. This phase is triggered by the reversal of ventricular volume change, i.e. $\frac{dV}{dt} > 0$. Symmetrically with phase (2), here the pressure is allowed to decrease while the volume is kept constant.

(6) *Passive filling*. This phase is triggered when ventricular pressure is lower than a threshold pressure $P_{post}$. During this phase the myocyte active tension is allowed to return to resting state and the volume is allowed to return to the initialised value $V_0$ through: $dP = -\frac{1}{C_p}\left(V - V_0\right) - \frac{1}{C_v} \cdot \frac{dV}{dt}$

where ventricular volumes were calculated at each time step using the divergence theorem (*Levrero-Florencio et al., 2020*). Here, both the $C_p$ and $C_v$ parameters are user defined and have been selected to allow full recovery of the initialised volume before the end of the cycle length.

**Appendix 1—table 3.** Parameters for boundary conditions and phase control at resting heart rate and fast pacing (in brackets).

| Name | Parameter | LV | RV | Unit |
|---|---|---|---|---|
| Pericardial stiffness | $K_{epi}$ | 10000 | | Ba cm$^{-1}$ |
| Time to initial pressure | $t_0$ | 0.02 | 0.02 | s |
| Initial pressure | $P_0$ | 5000 | 5000 | Ba |
| Duration of passive diastolic filling | $t_{diastole}$ | 0.08 (0.03) | 0.08 (0.03) | s |
| Pressure at end of diastole | $P_{endd}$ | 15000 | 15000 | Ba |
| Arterial compliance | C | 0.00055908 | 0.00055908 | cm$^3$ Ba$^{-1}$ |
| Arterial resistance | R | 250 | 100 | Ba s cm$^{-3}$ |
| Aortic pressure | $P_{art0}$ | 90000 | 20000 | Ba |
| Pressure at end of isovolumetric relaxation | $P_{post}$ | 10000 | 10000 | Ba |
| Penalty parameters for isovolumetric contraction | $C_v$ | 1 | 1 | cm$^3$ s$^{-1}$ Ba$^{-1}$ |
| Penalty parameters for isovolumetric relaxation | $C_v$ | 0.2 | 0.2 | cm$^3$ s$^{-1}$ Ba$^{-1}$ |
| Penalty parameters for passive filling | $C_p, C_v$ | 0.1,0.3 | 0.1,0.3 | cm$^3$ Ba$^{-1}$, cm$^3$ s$^{-1}$ Ba$^{-1}$ |

## 3. ECG simulation and biomarker calculation

Simulated ECGs were evaluated at prescribed electrode locations on the torso surface using the pseudo-ECG method (*Mincholé et al., 2019*), assuming the torso is an infinite volume conductor. The signals are normalised in the precordial leads according to the maximum amplitude, and they are then normalised separately for the limb leads. This addresses the limitation of this method in not being able to faithfully representing absolute amplitudes (*Ogiermann et al., 2021*). Only the precordial leads were evaluated because the biomarkers calculated for these leads were the most reliable.

ECG biomarkers as listed in *Appendix 1—table 8* were evaluated using a Python code as follows:

1. Separate each beat using the pacing R-to-R interval.
2. Resample the signal to 1000 Hz.
3. For each beat, use the voltage at end of each beat to offset the signal.
4. Evaluate the first and second order derivative of the voltage signal, apply filter on the derivatives using python package scipy.signal.lfilter with filtering parameters tuned manually to achieve best results for steps (4), (5) and (6). Normalize the absolute value of the filtered signal using the maximum absolute value.
5. Identify beginning of QRS by searching from the beginning of signal and finding the first time at which the first derivative becomes higher than the threshold 0.01*max(V)/30.
6. Identify the end of QRS manually.
7. Identify the end of T wave by searching from the end of signal and finding the first time point at which the first derivative becomes higher than the threshold 0.01*max(V)/30.
8. Isolate the signal segment from the end of QRS to the end of T wave and evaluate the peak absolute value to find T-wave peak and identify its timing.
9. Isolate the signal segment from the end of QRS to the peak of T wave. Evaluate mean dV using a window width of 3, then, beginning at the end of the QRS, the onset of T wave is identified as the time step at which the second derivative of voltage first becomes larger than 0.12*max(V)/30.
10. Evaluate the required biomarkers using the landmark points identified.

An example of the output of this delineation method is as below, showing QRS duration, QT duration, and T-wave duration (units ms) are shown after the lead name. Due to low reliability of this method in delineating the QRS complex, only the QT duration and T wave characteristics are reported in the main manuscript.

## 4. Border zone (BZ), remote zone (RZ) and scar ionic remodeling and their effects on AP and CaT biomarkers

For acute post-MI (within a week post-occlusion), three types of BZ remodelling (Acute BZ1-3) were considered based on previous modelling work and experimental canine data collected within 5 d post-MI (*Hund et al., 2008*; *Decker and Rudy, 2010*; *Arevalo et al., 2016*; *Tomek et al., 2017*). For chronic post-MI, ionic current remodelling measured from minipigs 5 mo post-MI with heart failure were used to generate Chronic BZ (affecting only the BZ) and Chronic RZ1 (also affecting the remote myocardium). Another type of remodelling, Chronic RZ2, was established based on multiple experimental data from failing human cardiomyocytes (*Schwinger et al., 1999*; *Jiang et al., 2002*; *Li et al., 2004*; *Zicha et al., 2004*; *Valdivia et al., 2005*; *Maltsev et al., 2007*; *Holzem Katherine et al., 2011*; *Chang et al., 2013*; *Elshrif and Cherry, 2014*; *Gomez et al., 2014*; *Hegyi et al., 2018*; *Høydal et al., 2018*). Furthermore, reduction of sodium current (*Valdivia et al., 2005*) and SERCA (*Jiang et al., 2002*), with enhanced CaMKII activity and slower calcium release (*Hoch et al., 1999*; *Maier and Bers, 2007*) were also implemented in Chronic BZ, RZ1 and RZ2, as observed in human failing cardiomyocytes. The remodelling in the infarcted scar region was modelled as previously published (*Wang et al., 2021*), which generated prolonged action potential than the normal zone as observed in the activation recovery interval (ARI) data of post infarction human and pigs (*Vaseghi et al., 2017*; *Srinivasan et al., 2019*).

**Appendix 1—table 4.** Ionic current remodelling for the acute and chronic stage BZ and RZs.

| Scaling | Acute BZ1 | Acute BZ2 | Acute BZ3 | Chronic BZ | Chronic RZ1 | Chronic RZ2 | Infarct |
|---|---|---|---|---|---|---|---|
| $G_{Na}$ | 0.4 *Hund et al., 2008*; *Decker and Rudy, 2010* | 0.38 *Arevalo et al., 2016* | 0.4 *Tomek et al., 2017* | 0.43 *Valdivia et al., 2005* | 0.43 *Valdivia et al., 2005* | 0.43 *Valdivia et al., 2005* | 0.4 |
| $G_{NaL}$ | | | | 1.275 *Hegyi et al., 2018* | 1.413 *Hegyi et al., 2018* | 2 *Valdivia et al., 2005*; *Maltsev et al., 2007* | |
| $G_{to}$ | 0.1 *Hund et al., 2008*; *Decker and Rudy, 2010* | | 0 *Tomek et al., 2017* | | | 0.6 *Beuckelmann et al., 1993*; *Li et al., 2004* | 0 |
| $G_{CaL}$ | 0.64 *Hund et al., 2008*; *Decker and Rudy, 2010* | 0.31 *Arevalo et al., 2016* | 0.64 *Tomek et al., 2017* | 0.7 *Hegyi et al., 2018* | | | 0.64 |
| $G_{Kr}$ | 0.7 *Hund et al., 2008*; *Decker and Rudy, 2010* | 0.3 *Arevalo et al., 2016* | | 0.89 *Hegyi et al., 2018* | 0.87 *Hegyi et al., 2018* | 0.6 *Ambrosi et al., 2013* | 0.7 |
| $G_{Ks}$ | 0.2 *Hund et al., 2008*; *Decker and Rudy, 2010* | 0.2 *Arevalo et al., 2016* | | | | 0.4 *Li et al., 2004* | |
| $G_{K1}$ | 0.3 *Hund et al., 2008*; *Decker and Rudy, 2010* | | 0.6 *Tomek et al., 2017* | 0.76 *Hegyi et al., 2018* | | 0.6 *Beuckelmann et al., 1993*; *Li et al., 2004* | 0.6 |
| $G_{NaK}$ | | | | | | 0.6 *Schwinger et al., 1999* | |
| $P_{Jup}$ | | | | 0.4 *Jiang et al., 2002*; *Høydal et al., 2018* | 0.4 *Jiang et al., 2002*; *Høydal et al., 2018* | 0.3 *Jiang et al., 2002*; *Høydal et al., 2018* | |
| $G_{KCa}$ | | | | 2 *Hegyi et al., 2018* | 2 *Hegyi et al., 2018* | 3.75 *Chang et al., 2013* | |
| $G_{ClCa}$ | | | | 1.25 *Hegyi et al., 2018* | 1.25 *Hegyi et al., 2018* | 1.25 *Hegyi et al., 2018* | |
| aCaMK | | | 1.5 *Tomek et al., 2017* | 1.5 *Hoch et al., 1999*; *Hund et al., 2008* | 1.5 *Hoch et al., 1999*; *Hund et al., 2008* | 1.5 *Hoch et al., 1999*; *Hund et al., 2008* | 1.5 |
| $Tau_{relp}$ | | | 6 *Maier et al., 2003* | 6 *Maier et al., 2003* | 6 *Maier et al., 2003* | 6 *Maier et al., 2003* | 6 |
| $G_{Cab}$ | | | 1.33 *Tomek et al., 2017* | | | | 1.33 |

**Appendix 1—table 5.** Comparison of the simulated AP, CaT and active tension (Ta) biomarkers with post myocardial infarction (MI) acute stage canine experimental data and chronic stage human experimental data.

| | Biomarkers | Experimental values | Simulated values |
|---|---|---|---|
| Acute Post-MI Stage | Canine epi NZ APD (ms) (mean ± SD) | 295±34 *Lue and Boyden, 1992* 210±15 *Gardner et al., 1985* 219±39 *Spear et al., 1983* Overall: [180, 329] | 227 |
| | Canine epi BZ APD (ms) (mean ± SD) | 346±60 *Lue and Boyden, 1992* 170±15 *Gardner et al., 1985* 220±26 *Spear et al., 1983* Overall: [194, 406] | BZ1: 284, BZ2: 256, BZ3: 208 |
| | Canine epi BZ Systolic Cai EBZ/NZ (%) | 74% *Licata et al., 1997* | NZ: 686, BZ1: 517 (75%), BZ2: 133 (20%), BZ3: 457 (67%), |
| | Canine epi BZ Voltage Clamp Cai at 0 mV EBZ/NZ | 53% *Pu et al., 2000* | |
| | Canine Cell shortening EBZ/NZ % | 12% *Licata et al., 1997* | NZ systolic Ta: 40, BZ1: 24 (60%), BZ2: 0.37 (1%), BZ3: 20 (50%) |
| Chronic Post-MI Stage | Human Mid Systolic Cai Failing/Non-Failing (%) | 49% *Piacentino et al., 2003* | NZ (800 ms CL):1219, RZ1: 744 (60%), RZ2: 608 (50%) |
| | Human Mid Systolic Cai MI/normal (%) 1 Hz | 37.5% *Høydal et al., 2018* | |
| | Human Mid Diastolic Cai Failing/Non-Failing (%) | 96% *Piacentino et al., 2003* | NZ (800 ms CL): 80, RZ1: 37 (46%), RZ2: 39 (49%) NZ (500 ms CL): 86, RZ1: 52 (61%), RZ2: 62 (72%) |
| | Human Mid Diastolic Cai MI/normal (%) 1 Hz | 115% *Høydal et al., 2018* | |
| | Human Mid Cell shortening MI/normal (%) 1 Hz | 33% *Høydal et al., 2018* | NZ systolic Ta (800 ms CL): 65, RZ1: 58 (89%), RZ2: 45 (70%) |

**Appendix 1—table 6.** Simulated AP, CaT and Ta biomarkers from baseline NZ, BZ and RZ epi-, mid-, and endocardial single cell models.

For the acute stage, the Acute BZ1 and BZ2 induced significant APD prolongation, while the BZ3 led to mild APD shortening. The Acute BZ1 and BZ3 had similar degree of reduction in systolic Ca and Ta, whereas the Acute BZ2 had more severe loss of contractility. For the chronic stage, the Chronic RZ1 and RZ2 had similar decrease in systolic Ca and Ta than control, but the RZ2 had more severe APD prolongation than the Chronic RZ1.

| Type | APD90 (ms) | CaTD90 (ms) | Diastolic Ca (nM) | Systolic Ca (nM) | Diastolic Ta (kPa) | Systolic Ta (kPa) |
|---|---|---|---|---|---|---|
| Control (800ms) | epi: 227, mid: 336, endo: 263 | epi: 298, mid: 333, endo: 336 | epi: 62.32, mid: 79.74, endo: 70.70 | epi: 686.27, mid: 1218.83, endo: 477.71 | epi: 0.06, mid: 0.10, endo: 0.07 | epi: 40.00, mid: 65.47, endo: 23.87 |
| Acute BZ1 | epi: 284, mid: 391, endo: 315 | epi: 289, mid: 339, endo: 325 | epi: 59.63, mid: 63.52, endo: 65.62 | epi: 517.13, mid: 691.30, endo: 342.98 | epi: 0.05, mid: 0.06, endo: 0.06 | epi: 24.33, mid: 44.78, endo: 10.46 |
| Acute BZ2 | epi: 256, mid: 373, endo: 341 | epi: 266, mid: 342, endo: 334 | epi: 45.11, mid: 57.08, endo: 57.98 | epi: 133.09, mid: 326.57, endo: 184.93 | epi: 0.03, mid: 0.05, endo: 0.05 | epi: 0.37, mid: 8.97, endo: 1.48 |
| Acute BZ3 | epi: 208, mid: 316, endo: 247 | epi: 256, mid: 295, endo: 288 | epi: 49.05, mid: 60.49, endo: 59.79 | epi: 457.49, mid: 834.58, endo: 352.71 | epi: 0.03, mid: 0.05, endo: 0.05 | epi: 19.66, mid: 52.08, endo: 11.18 |

*Appendix 1—table 6 Continued on next page*

*Appendix 1—table 6 Continued*

| Type | APD90 (ms) | CaTD90 (ms) | Diastolic Ca (nM) | Systolic Ca (nM) | Diastolic Ta (kPa) | Systolic Ta (kPa) |
|---|---|---|---|---|---|---|
| Chronic BZ | epi: 235, mid: 362, endo: 293 | epi: 419, mid: 444, endo: 474 | epi: 41.90, mid: 40.80, endo: 50.59 | epi: 324.87, mid: 499.97, endo: 285.67 | epi: 0.03, mid: 0.03, endo: 0.04 | epi: 10.44, mid: 31.40, endo: 7.76 |
| Chronic RZ1 | epi: 247, mid: 411, endo: 313 | epi: 426, mid: 462, endo: 478 | epi: 39.64, mid: 37.47, endo: 49.34 | epi: 459.72, mid: 744.07, endo: 387.91 | epi: 0.03, mid: 0.05, endo: 0.04 | epi: 25.67, mid: 57.56, endo: 18.40 |
| Chronic RZ2 | epi: 392, mid: 591, endo: 467 | epi: 498, mid: 569, endo: 557 | epi: 39.77, mid: 38.61, endo: 49.51 | epi: 444.67, mid: 607.59, endo: 361.11 | epi: 0.03, mid: 0.09, endo: 0.05 | epi: 25.14, mid: 44.80, endo: 16.17 |
| Acute and (chronic) scar | epi: 250, mid: 366, endo: 295 | epi: 261, mid: 304, endo: 296 | epi: 49.17, mid: 62.49, endo: 59.70 | epi: 502.85, mid: 883.02, endo: 375.70 | epi: 0.03 (0), mid: 0.06 (0), endo: 0.05 (0) | epi: 24.23 (0), mid: 53.21 (0), endo: 13.28 (0) |
| Control (500ms) | epi: 210, mid: 306, endo: 240 | epi: 265, mid: 276, endo: 298 | epi: 58.61, mid: 85.60, endo: 67.91 | epi: 760.33, mid: 1740.38, endo: 531.49 | epi: 0.38, mid: 1.81, endo: 0.33 | epi: 42.77, mid: 68.04, endo: 27.54 |
| RZ1 (500ms) | epi: 226, mid: 296, endo: 271 | epi: 370, mid: 384, endo: 396 | epi: 47.10, mid: 52.52, endo: 65.07 | epi: 470.50, mid: 712.75, endo: 385.73 | epi: 0.50, mid: 2.04, endo: 0.51 | epi: 25.84, mid: 51.69, endo: 17.28 |
| RZ2 (500ms) | epi: 316, mid: 395, endo: 366 | epi: 389, mid: 401, endo: 410 | epi: 50.88, mid: 61.53, endo: 65.99 | epi: 425.66, mid: 605.85, endo: 337.64 | epi: 0.52, mid: 1.94, endo: 0.46 | epi: 21.20, mid: 41.79, endo: 12.23 |

## 5. Validation using clinical and experimental data

The clinical ECGs in *Figure 1* of main manuscript were extracted from the following:

Acute BZ1 comparison (T-wave inversion): PTB Diagnostic ECG Database (*Goldberger et al., 2000*; *Bousseljot et al., 2009*), patient number 014 who had anterior infarction. The ECG was taken 10 days after infarction.

Acute BZ2 comparison (Brugada phenocopy): Extracted from *Figure 1B* of a clinical paper describing Brugada phenocopy (*Anselm et al., 2014*) and enhanced using bespoke python script. The patient had acute inferior ST segment elevation myocardial infarction with right ventricular involvement.

Acute BZ3 comparison (Normal ST-T): PTB Diagnostic ECG Database (*Goldberger et al., 2000*; *Bousseljot et al., 2009*), patient number 051 who had antero-septal infarction. The ECG was taken 10 d after infarction.

Chronic RZ1 comparison (Slight QT prolongation): PTB Diagnostic ECG Database (*Goldberger et al., 2000*; *Bousseljot et al., 2009*), patient number 042 who had antero-septal infarction. The ECG was taken at ~20 mo follow up.

Chronic RZ2 comparison (Large T-wave): PTB Diagnostic ECG Database (*Goldberger et al., 2000*; *Bousseljot et al., 2009*), patient number 033 who had antero-septal infarction. The ECG was taken 3 mo after infarction.

**Appendix 1—table 7.** Comparison of the ECG and mechanical biomarkers from biventricular electromechanical simulations against literature values at resting heart rate.

QTc was calculated using Bazett's formula from the simulated QT intervals. Post-MI RVEF values were from ST-segment elevation myocardial infarction patients whose culprit and chronic total occlusion were not in the right coronary artery. VA: ventricular arrhythmia; VT: ventricular tachycardia; SDB: sleep disordered breathing. Our simulated ECG and mechanical biomarker values are mostly consistent with the clinically reported biomarker ranges.

| Biomarkers | Control | | Acute Stage Post-MI | | Chronic Stage Post-MI | |
|---|---|---|---|---|---|---|
| Electrophysiological Biomarkers | Literature | Simulation | Literature | Simulation | Literature | Simulation |

*Appendix 1—table 7 Continued on next page*

*Appendix 1—table 7 Continued*

| Biomarkers | Control | | Acute Stage Post-MI | | Chronic Stage Post-MI | |
|---|---|---|---|---|---|---|
| QRS duration (ms) | 96 ± 9 in men, 85 ± 6 in women *Carlsson et al., 2006* | 79 ± 2 | 88 ± 35 *Yerra et al., 2006* | 91±5, 95±9, 92±6 | Max 127 ± 16 without VT Min 81 ± 15 without VT *Perkiömäki et al., 1995* Max 137 ± 25 with VT Min 89 ± 20 with VT *Perkiömäki et al., 1995* | 94 ± 6, 93 ± 5 |
| QTc interval (Bazett formula) (ms) | 350–440 *Johnson and Ackerman, 2009* | 360 ± 1 | 423 ± 50 without VA *Ahnve, 1985* 460±40 with VA *Ahnve, 1985* | 398 ± 27, 415 ± 4, 376 ± 5 | Max 448 ± 39 without VT Min 383 ± 20 without VT *Perkiömäki et al., 1995* Max 493 ± 51 with VT Min 388 ± 30 with VT ve stiffness parameters were calibrated based on *Perkiömäki et al., 1995* | 430 ± 4, 578 ± 3 |
| Mechanical Biomarkers | Literature | Simulation | Literature | Simulation | Literature | Simulation |
| LVEDV (mL) | 142 ± 21 (SSFP-CMR) *Maceira et al., 2006a* | 129 | 116 ± 15 *Uslu et al., 2013* | 124–125 | 106 ± 12 *Uslu et al., 2013* | 126 |
| RVEDV (mL) | 144 ± 23 (SSFP-CMR) *Maceira et al., 2006b* | 131 | 129 ± 28 with SDB *Buchner et al., 2015* 132 ± 28 without SDB *Buchner et al., 2015* | 131 | 143 ± 29 with SDB *Buchner et al., 2015* 132 ± 31 without SDB *Buchner et al., 2015* | 133 |
| LVESV (mL) | 47 ± 10 (SSFP-CMR) *Maceira et al., 2006a* | 60 | 61 ± 12 *Uslu et al., 2013* | 65~72 | 52 ± 10 *Uslu et al., 2013* | 65 |
| RVESV (mL) | 50±14 (SSFP-CMR) *Maceira et al., 2006b* | 63 | 56 21 with SDB *Buchner et al., 2015* 53 ± 16 without SDB *Buchner et al., 2015* | 64 | 58 ± 21 with SDB *Buchner et al., 2015* 51 ± 15 without SDB *Buchner et al., 2015* | 64 |
| LVEF (%) | 67 ± 4.6 (SSFP-CMR) *Maceira et al., 2006a*, 62±7 (RNV) *Nemerovski et al., 1982* | 53 | 48 ± 8 *Uslu et al., 2013* | 43~47 | 52 ± 7 *Uslu et al., 2013* | 48 |
| RVEF (%) | 48 ± 5 (RNV) *Nemerovski et al., 1982* | 52 | 53.0 ± 7.1 *van Veelen et al., 2022* | 51 | 55.9 ± 5.4 *van Veelen et al., 2022* | 52 |

## 6. Sensitivity analysis of ECG and LVEF to electrophysiological heterogeneities, calcium sensitivity, and sheet active tension

We performed the following sets of electromechanical simulations varying the apex-to-base and transmural heterogeneities, calcium sensitivity, and sheet active tension, in order to evaluate the effect of these on ECG morphology and LVEF.

The baseline apex-to-base gradient is evaluated by scaling the conductance of the slow delayed rectifier potassium current (GKs) according to:

$$sf = 0.2^{z_{scale}}$$

$$z_{scale} = 2z - 1$$

where z is the normalised longitudinal coordinate that is equal to zero at the apex and one at the base. This gives a scaling factor of 5 at the apex and 0.2 at the base (i.e., [0.2, 5]). We vary the magnitude

of this gradient by changing the basis of exponential for the sf equation through the values [0.1, 0.2, 0.3], to achieve ranges of [0.1, 10], [0.2, 5], [0.3, 3.3] for the scaling factor (*Appendix 1—figure 3A*). In addition, we also reverse the gradient direction thus:

$$z_{scale} = -2z + 1$$

The effect of changes in range of the apex-to-base gradient is negligible in LVEF (*Appendix 1—figure 3B*) and ECG morphology (*Appendix 1—figure 3C*) except in leads V4 and V5, where the amplitude of the T-wave decreased with decreasing range of GKs scaling. A complete reversal of the gradient caused a more dramatic decrease in T-wave amplitude and reduction in T-wave duration apparent in V3 to V5 (*Appendix 1—figure 3C*, red line), with no changes in LVEF.

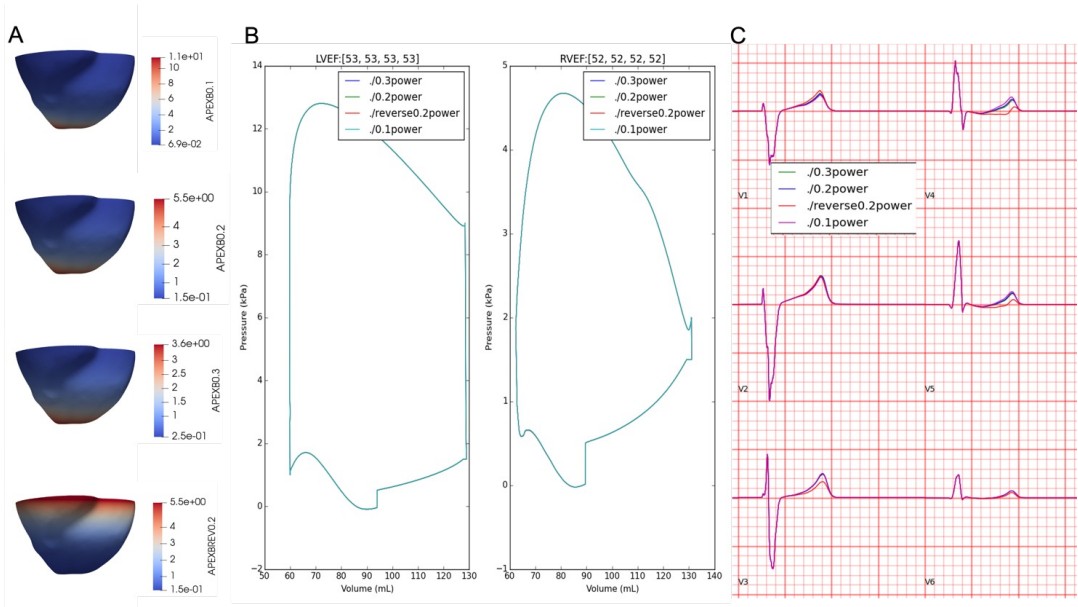

**Appendix 1—figure 3.** Effects of apex-to-base gradient (**A**) on pressure-volume, LVEF (**B**), and ECG morphology (**C**).

The baseline transmural heterogeneity has a 30%, 40%, and 30% split of endocardial, mid-myocardial, and epicardial cell types across the wall (*Appendix 1—figure 4A*). To modify this, we created two other transmural compositions where the mid-myocardial layer is removed, and the endo- vs. epicardial split is varied from 30%–70% to 50%–50% (*Appendix 1—figure 4A*). The removal of the mid-myocardial layer caused an increase in end systolic volume, a reduction in LVEF, and a decreased T-wave amplitude and increased QT interval across all precordial leads (compare green and blue). Increasing the proportion of epicardial cell type (compare red and blue) caused a reduction in T-wave amplitude and an increase in QT interval. The polarity of the T-wave remained unchanged.

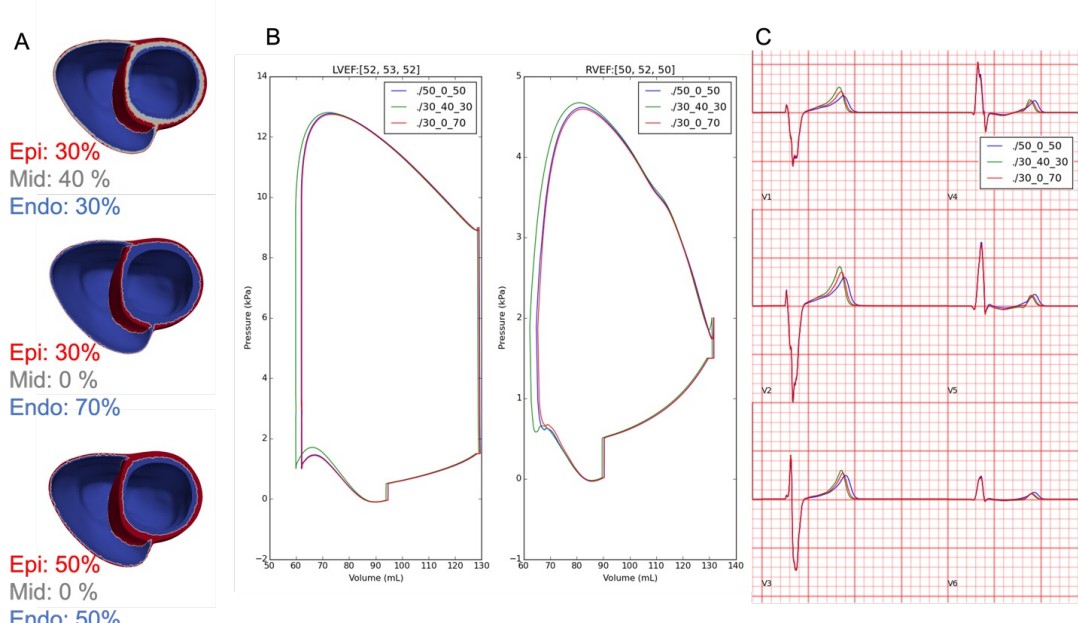

**Appendix 1—figure 4.** Effects of transmural electrophysiological heterogeneity (**A**) on pressure-volume, LVEF (**B**), and ECG morphology (**C**).

Additionally, we altered the calcium sensitivity of troponin binding in the excitation-contraction coupling (Ca50 parameter from the baseline value *Land et al., 2017*) of 0.805 $\mu M$ to a range of values: [0.5, 0.7, 1.0]. This had no effect on the ECG (*Appendix 1—figure 5A*). With increasing Ca50 values, there was an increase in end diastolic volume and end systolic volume, resulting in negligible changes to the LVEF, except in the case of 0.5, where the LVEF is reduced by 2% (*Appendix 1— figure 5B*).

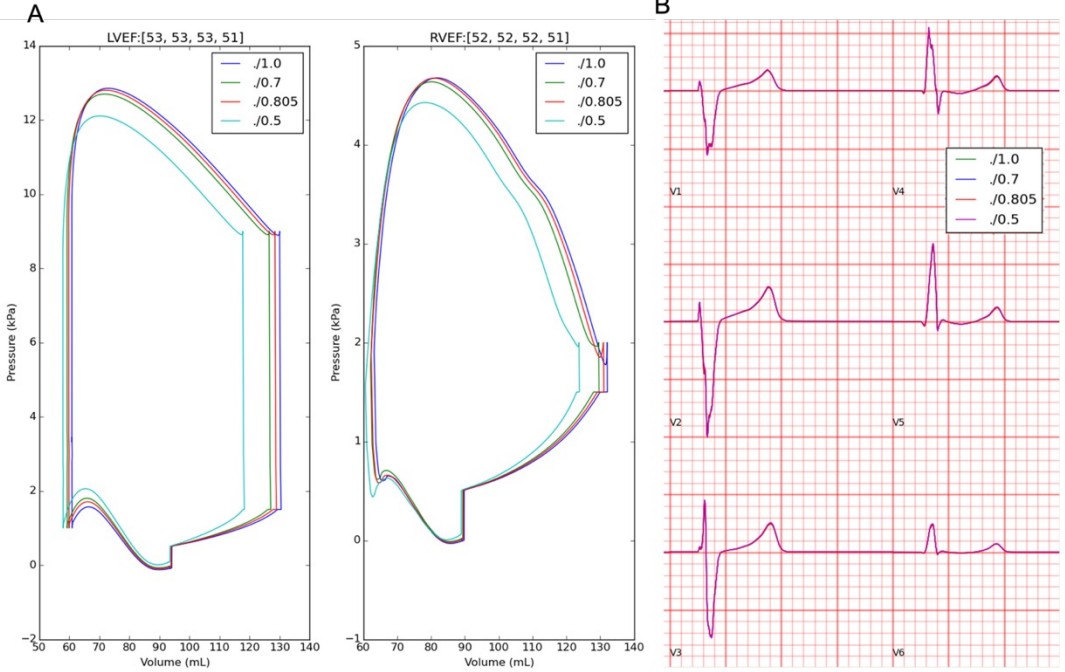

**Appendix 1—figure 5.** Effects of troponin calcium sensitivity on pressure-volume, LVEF (**A**), and ECG morphology (**B**).

Finally, we also altered the sheet-direction active tension from the baseline, where it is 30% of the fibre active tension, to a range of values: [0%, 50%, 60%]. With increasing percentage sheet activation LVEF increased (*Appendix 1—figure 6A*) with negligible effect on ECG morphology (*Appendix 1—figure 6B*). Sheet activation above and including 50% caused numerical instabilities that caused the simulation to terminate prematurely during the isovolumic contraction phase of the cardiac cycle.

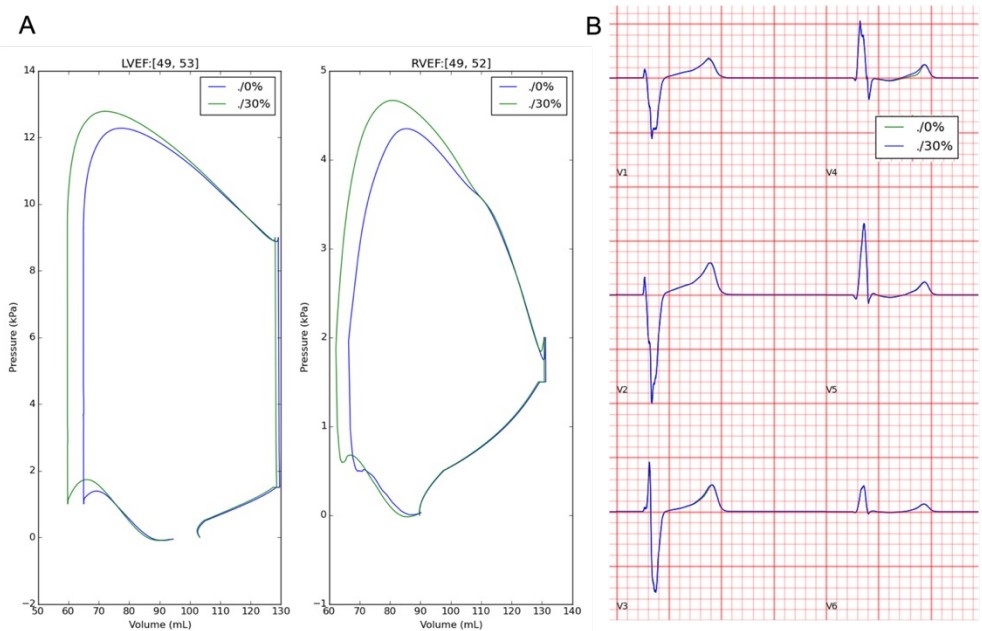

**Appendix 1—figure 6.** Effects of sheet activation percentage on pressure-volume, LVEF (**A**), and ECG morphology (**B**).

## 7. Biventricular electromechanical simulations of acute and chronic post-MI, ECG and pressure volume characteristics

Additional results of simulations for the acute and chronic post-MI phenotypes:

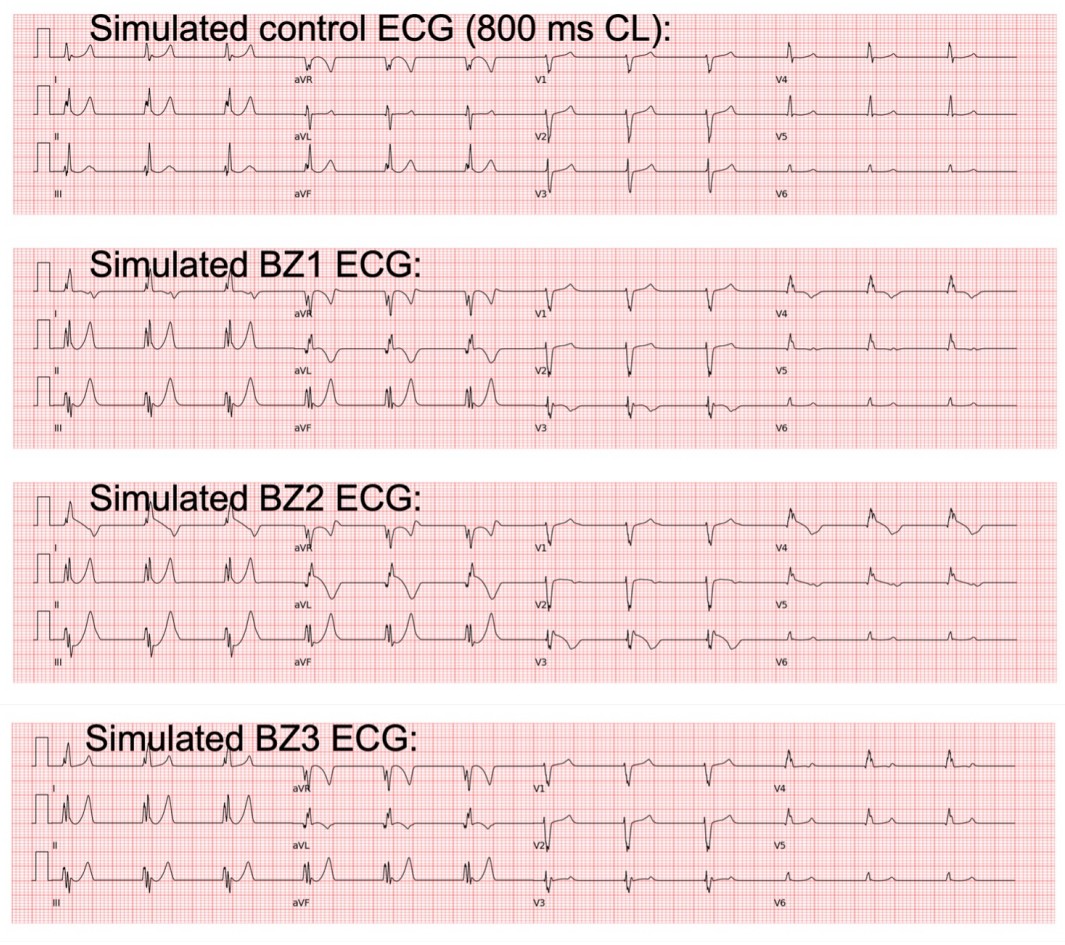

**Appendix 1—figure 7.** Simulated acute stage 12-lead ECGs for Acute BZ1-3. Acute BZ1 caused T wave inversion in precordial leads of V3 and V4, where the QT prolongation was more significant. Acute BZ2 caused Brugada phenocopy in leads V3-V5, while Acute BZ3 produced similar ECG morphology as the control case.

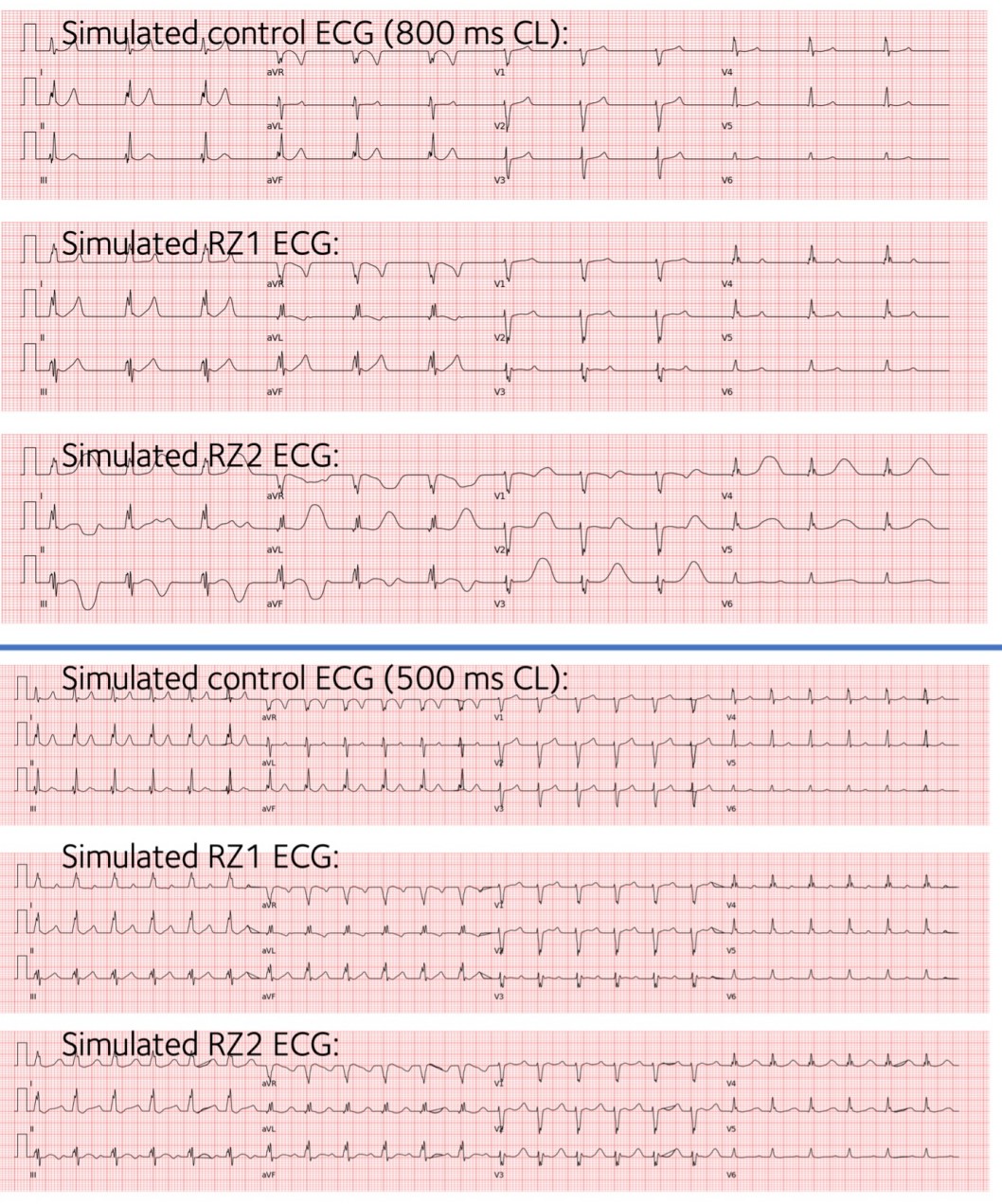

**Appendix 1—figure 8.** Simulated chronic stage 12-lead ECGs for Chronic RZ1 and Chronic RZ2, both combined with Chronic BZ. Both produced normal ECG morphology, and T waves are wider and taller in the anterior leads (V2–V4) of Chronic RZ2.

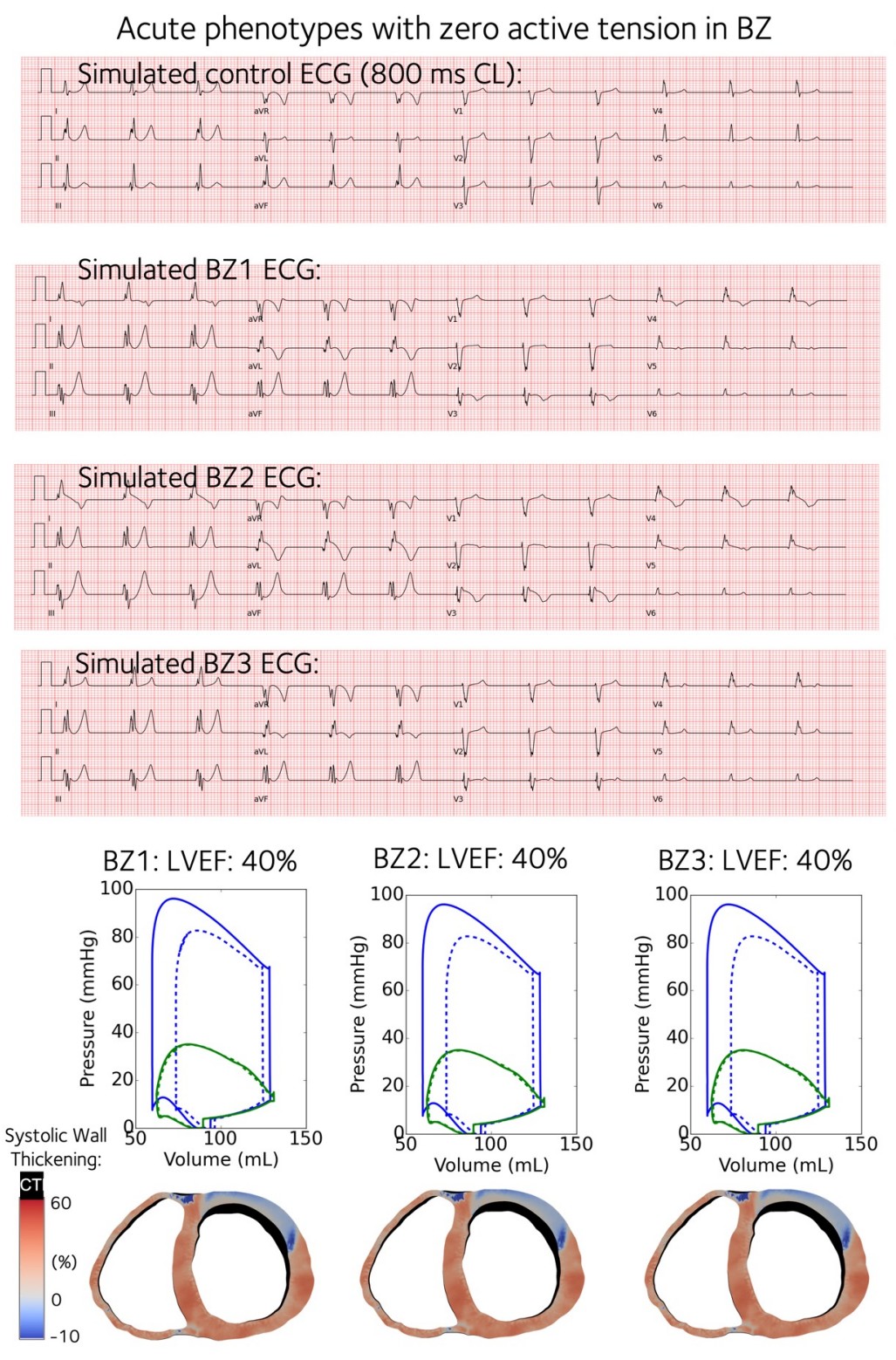

**Appendix 1—figure 9.** Simulated acute stage 12-lead ECGs for Acute BZ1-3 with contractility turned off in the BZs. Acute BZ1 caused T wave inversion in precordial leads of V3 and V4, where the QT prolongation was more significant. Acute BZ2 caused Brugada phenocopy in leads V3-V5, while Acute BZ3 produced similar ECG morphology as the control case.

**Appendix 1—table 8.** Simulated ECG biomarkers from biventricular electromechanical simulations for the acute and the chronic post-MI stages.

For the acute stage, Acute BZ1 and BZ2 caused significant QT prolongation, longer T peak to T end, whereas the Acute BZ3 induced milder effects. For the chronic stage, both Chronic RZ1 and RZ2 led to QT prolongation, with RZ2 also generating longer T wave duration, T peak to T end, and T start to T peak than control and RZ1 at CL = 800ms. At fast pacing of CL = 500ms, both Chronic RZ1 and RZ2 caused longer QT, T wave and T start to T peak durations. For both stages, the QT dispersions did not reflect the repolarization dispersion very well.

| ECG biomarkers | Control | Acute BZ1 | Acute BZ2 | Acute BZ3 | Chronic RZ1 | Chronic RZ2 | Control CL = 500ms | Chronic RZ1 CL = 500ms | Chronic RZ2 CL = 500ms |
|---|---|---|---|---|---|---|---|---|---|
| QRS duration (ms) | 79±2 | 91±5 | 95±9 | 92±6 | 94±6 | 93±5 | 86±7 | 86±7 | 86±7 |
| T duration (ms) | 96±17 | 163±40 | - | 113±18 | 122±42 | 291±20 | 97±19 | 100±15 | 140±37 |
| T peak to T end (ms) | 59±10 | 101±31 | 122±46 | 68±8 | 68±11 | 153±30 | 57±8 | 55±12 | 82±17 |
| T start to T peak (ms) | 38±8 | 61±30 | - | 45±10 | 54±32 | 138±12 | 40±11 | 45±6 | 58±22 |
| QT interval (ms) | 322±1 | 356±24 | 371±3 | 336±5 | 385±4 | 578±3 | 305±2 | 344±4 | 419±4 |
| QT dispersion (precordial) (ms) | 3 | 55 | 9 | 7 | 7 | 7 | 4 | 6 | 5 |

## 8. Electrophysiological characteristics of post-MI in more detail

The simulated transmural gradients of activation times, repolarisation times, and action potential duration (APD90) are shown in more detail below. Despite the inclusion of the mid-myocardial cell type that has elevated APD90 in the biventricular model, the transmural profile of APD90 in the simulated APD90 map is monotonically decreasing from endocardium to epicardium due to electronic coupling effects.

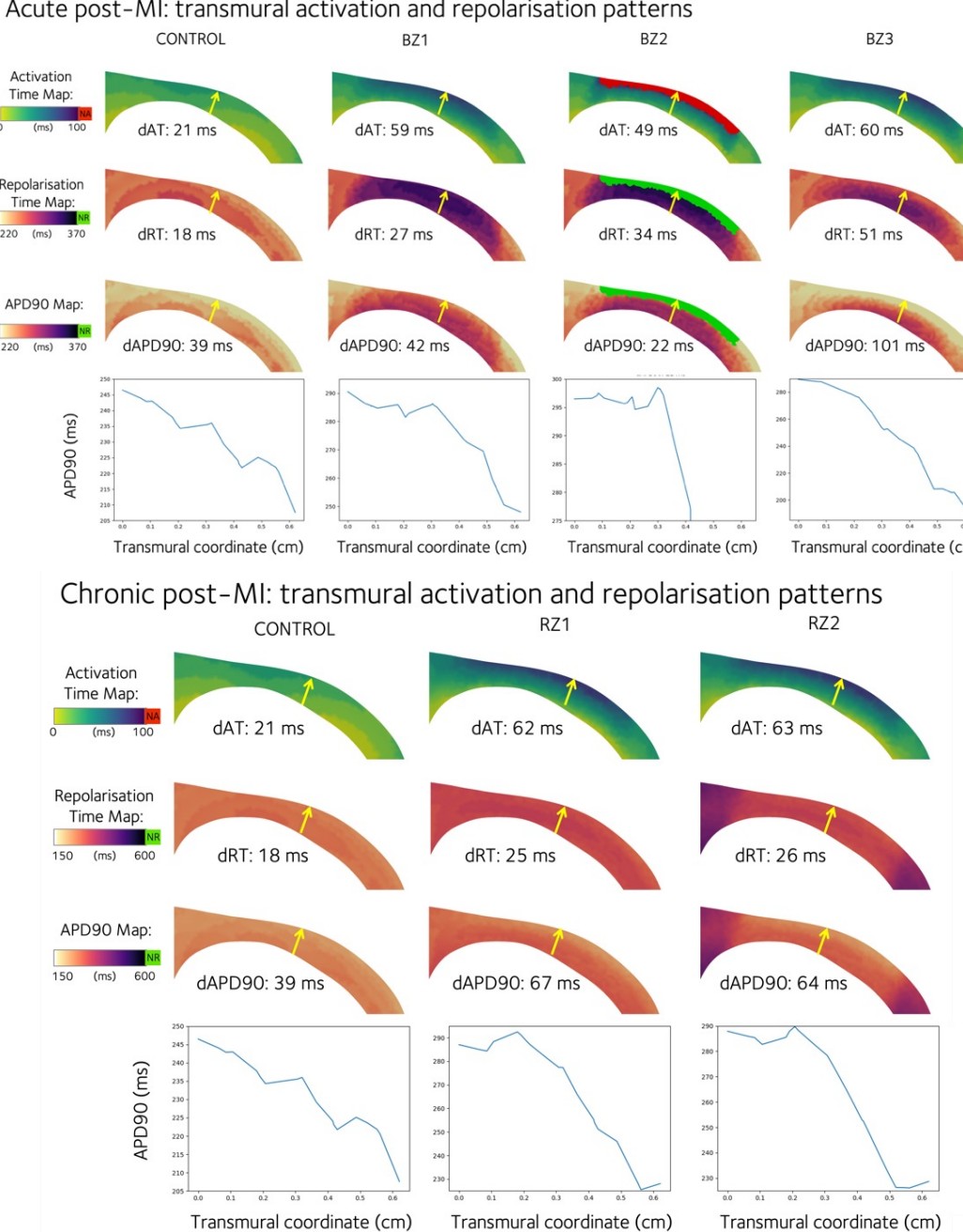

**Appendix 1—figure 10.** Transmural activation time (AT), repolarisation time (RT), and action potential duration at 90% repolarisation (APD90) are shown for the acute (top panel) and chronic (bottom panel) phenotypes. A mid-ventricular anterior transmural slice is taken from the left ventricle that shows the cross-section of the anterior infarction and border zones. The transmural gradient is evaluated as the quantity of interest at the epicardium minus that at the endocardium, and is given as dAT, dRT, and dAPD90 values for each cross-section, evaluated at the beginning and end of the yellow arrows. APD90 is plotted across a transmural line as indicated by the yellow arrow.

## 9. Explanation of the BZ2 activation pattern

Compared to the other cases, the acute BZ2 had the strongest inhibition of the L-type calcium current, and therefore the least safe epicardial conduction. Due to the transmural differences of the L-type calcium channel expression, the infarct zone in the midmyocardium had a higher level of the

calcium current than the epicardial BZ, which explained the more robust conduction in the infarct zone than in the BZ.

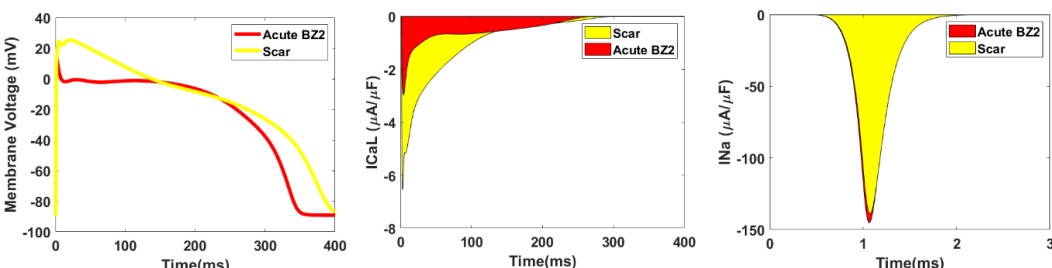

**Appendix 1—figure 11.** Single cell simulations of the action potential (left), L-type calcium ionic current (middle) and sodium ionic current (right) compared between the scar region and the acute BZ2 help to explain the activation pattern in BZ2.

**Appendix 1—table 9.** Simulated pressure-volume mechanical biomarkers for each heart beat from biventricular electromechanical simulations for acute and chronic stages post-MI: left and right end diastolic volumes (EDVL, EDVR), left and right stroke volumes (SVL, SVR), left and right ventricular ejection fractions (LVEF, RVEF).

For the acute stage, Acute BZ1 and Acute BZ3 generated the same degree of reduction in SVL and LVEF, whereas the Acute BZ2 induced the smallest SVL and LVEF. For the chronic stage, both Chronic RZ1 and Chronic RZ2 produced the same SVL and LVEF at both pacing rates despite their difference in the degree of repolarization heterogeneity.

| | Pressure-volume Biomarkers | Control | Acute BZ1 | Acute BZ2 | Acute BZ3 | Chronic RZ1 | Chronic RZ2 |
|---|---|---|---|---|---|---|---|
| | EDVL (mL) | 129, 129, 129 | 124,124,124 | 124,125,125 | 124,124,124 | 127, 126, 126 | 127, 126, 126 |
| | EDVR (mL) | 130,131,131 | 130,131,131 | 130,131,131 | 130,131,131 | 132,133,133 | 132,133,133 |
| | SVL (mL) | 68, 69, 69 | 59,59,59 | 53,53,53 | 59,59,59 | 62, 61, 61 | 62, 61, 61 |
| | SVR (mL) | 68,68,68 | 67,67,67 | 67,67,67 | 67,67,67 | 69,69,69 | 69,69,69 |
| | LVEF (%) | 53,53,53 | 47,47,47 | 43,43,43 | 47,47,47 | 49, 48, 48 | 49, 48, 48 |
| | RVEF (%) | 52,52,52 | 51,51,51 | 51,51,51 | 51,51,51 | 52,52,52 | 52,52,52 |
| | Peak left systolic pressure (kPa) | 12,12,12 | 11,11,11 | 11,11,11 | 11,11,11 | 11, 11, 11 | 11, 11, 11 |
| 800 ms CL | Peak right systolic pressure (kPa) | 4,4,4 | 4,4,4 | 4,4,4 | 4,4,4 | 4,4,4 | 4,4,4 |
| | EDVL (mL) | 111,112,113, 113,113,113 | | | | 114,107,111, 108,110,108 | 116,108,111, 109,110,109 |
| | EDVR (mL) | 123,122,123, 123,123,123 | | | | 125,117,122, 119,121,119 | 126,119,123, 120,121,120 |
| | SVL (mL) | 53,53,54, 54,54,54 | | | | 50,43,47, 44,45,44 | 51,43,46, 44,45,44 |
| | SVR (mL) | 61,61,61, 61,61,61 | | | | 64,56,62, 56,60,57 | 65,57,62, 59,60,59 |
| | LVEF (%) | 47,47,47, 47,47,47 | | | | 44,39,42, 40,41,40 | 44,39,41, 40,41,40 |
| | RVEF (%) | 49,49,49, 49,49,49 | | | | 51,47,50, 47,50,48 | 51,48,50, 49,49,49 |
| | Peak left systolic pressure (kPa) | 11,11,11, 11,11,11 | | | | 10,10,10, 10,10,10 | 10,10,10, 10,10,10 |
| 500 ms CL | Peak right systolic pressure (kPa) | 4,4,4, 4,4,4 | NA | | | 4,4,4, 4,4,4 | 4,4,4, 4,4,4 |

## 10. Post-MI SERCA and CaMKII remodeling promote alternans generations in models with more preserved calcium magnitudes

In our cellular population of post-MI models, higher inducibility of alternans were observed at both the acute and the chronic stages at fast pacing (*Appendix 1—figure 12*).

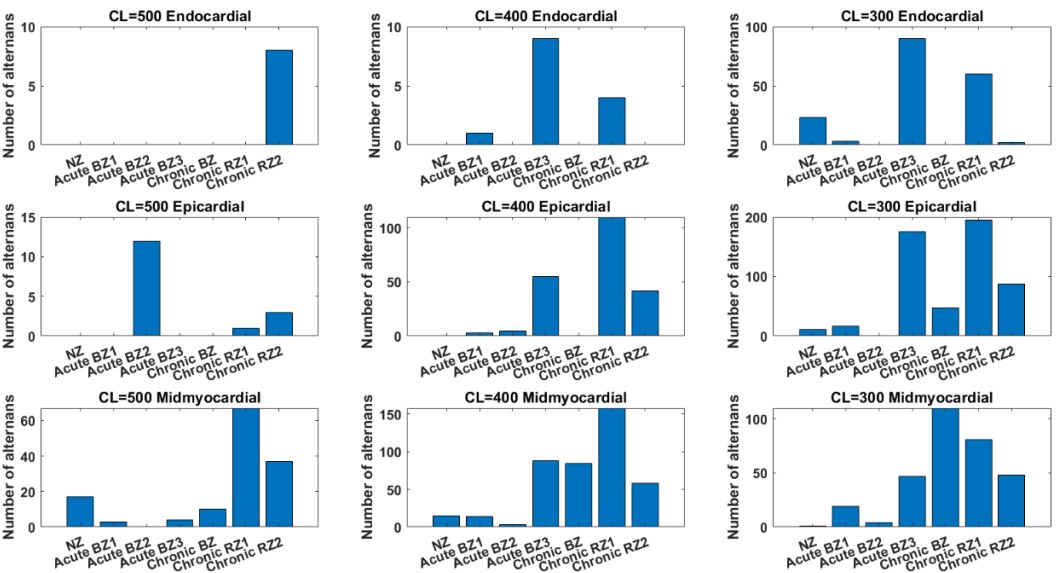

**Appendix 1—figure 12.** The effects of the BZ and RZ remodelling of the acute and chronic stages on alternans generation in the population of 245 population of models at CL = 500ms, 400ms and 300ms.

All three types of chronic post-MI remodeling promoted alternans generation, especially in the epicardial and midmyocardial populations (*Appendix 1—table 10*). Alternans in the midmyocardial layer were mostly due to EADs, whereas the epicardial alternans were repolarization alternans (*Appendix 1—figures 13 and 14*).

**Appendix 1—table 10.** Number of alternans induced by three chronic remodelling at CL = 500, 400 and 300ms in endocardial, midmyocardial and epicardial population of models.

| Population of models (n=245) | No. of alternans at CL = 300ms | No. of alternans at CL = 400ms | No. of alternans at CL = 500ms | Key parameters for alternans |
|---|---|---|---|---|
| Chronic BZ | Mid (110)>Epi (47) | Mid only (84) | Mid only (10) | $\uparrow G_{CaL}$, $\uparrow G_{Kr}$, $\uparrow P_{Jup}$ |
| Chronic RZ1 | Epi (195)>Mid (81)>Endo (60) | Mid (158)>Epi (110)>Endo (4) | Mid (67)>Epi (1) | $\uparrow G_{CaL}$, $\uparrow P_{Jup}$ |
| Chronic RZ2 | Epi (88)>Mid (48)>Endo (2) | Mid (58)>Epi (42) | Mid (37)>Endo (8)>Epi (3) | $\uparrow G_{CaL}$, $\uparrow G_{Kr}$, $\downarrow G_{NCX}$, $\uparrow P_{Jup}$, $\uparrow P_{Jrel}$ |

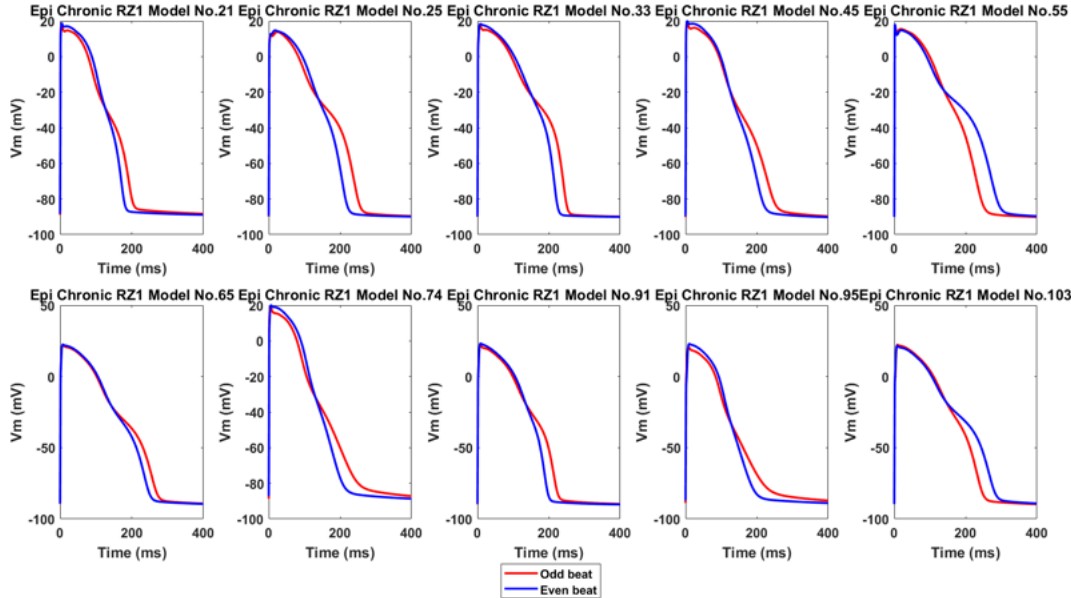

**Appendix 1—figure 13.** Ten representative alternans in the epicardial population of Chronic RZ1, showing calcium-driven repolarization alternans without EADs.

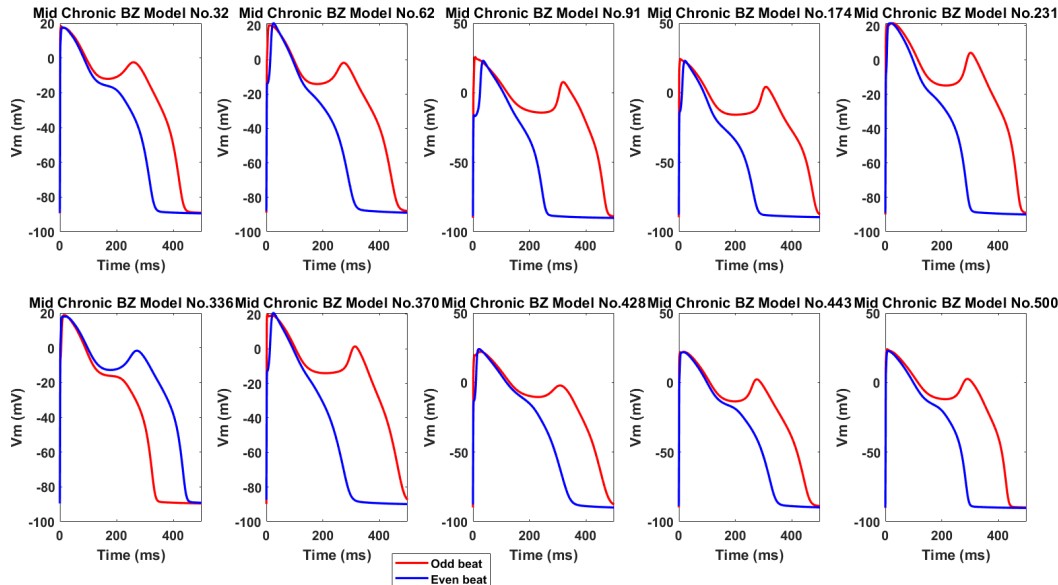

**Appendix 1—figure 14.** Ten representative alternans in the midmyocardial population of Chronic BZ, showing EAD as a major cause of big alternans.

Here we investigated: (1) what are the key individual post-MI ionic current remodeling contributing to the generation of alternans, and (2) what are the underlying ionic currents in the population of models that determine whether a post-MI model is prone for alternans induction?

In order to illustrate the underlying mechanisms, the baseline chronic remote zone model (with Chronic RZ1 remodeling) was chosen as an example in *Appendix 1—figure 15*. At CL = 300ms, the model had alternations of long and short APDs in the odd and even beats (blue solid traces), and it was clear that the alternans was associated with the insufficient calcium re-uptake and slow calcium recovery in junctional sarcoplasmic reticulum (JSR). When the inhibition of SERCA pump ($J_{up}$) was switched off (red dashed traces), alternans disappeared along with a significant increase of the

JSR calcium level. Apart from the insufficient calcium re-uptake in the remodeling, higher CaMKII activation and slower calcium release further contributed to the alternans. When CaMKII activation and $J_{rel}$ kinetics were switched back to normal (yellow dashed traces), the duration of calcium release was shorter, and there was more time for JSR calcium to recover before the next beat, leading to the elimination of alternans (yellow dashed traces).

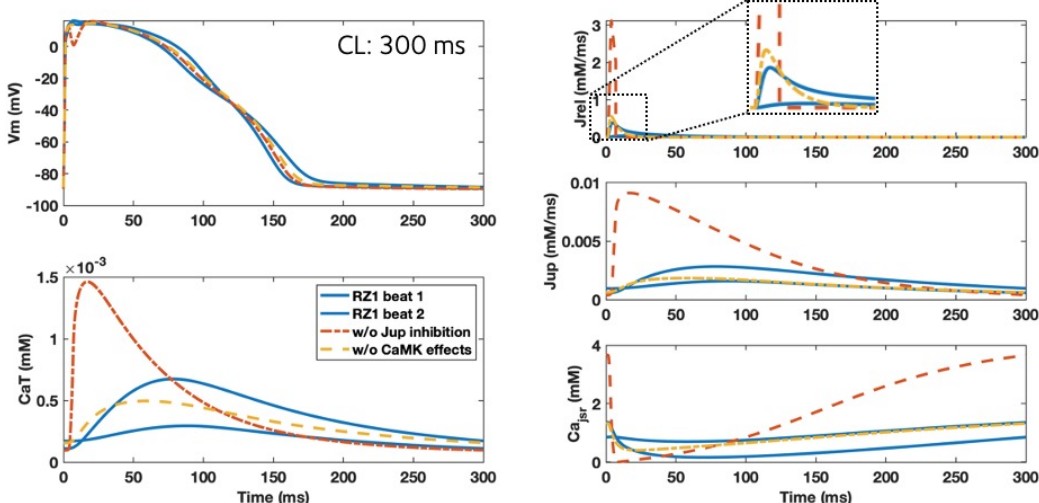

**Appendix 1—figure 15.** Inhibition of $J_{up}$ and slower calcium release ($J_{rel}$) caused by enhanced CaMKII activity promoted alternans.

On the other hand, $I_{KCa}$ augmentation played a protective role against alternans generation by shortening APD and therefore regulating calcium dynamics (*Appendix 1—figures 16 and 17*):

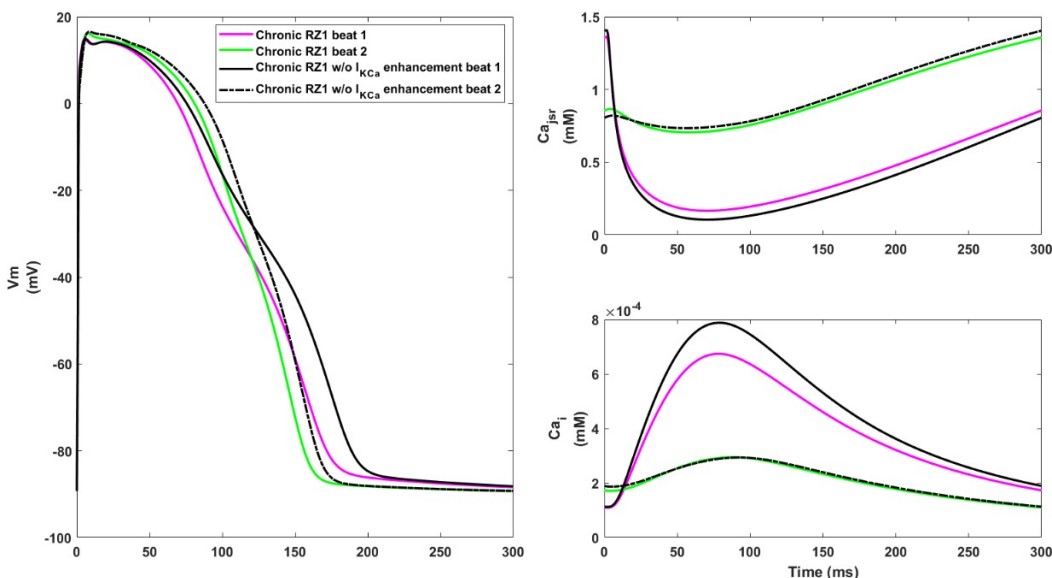

**Appendix 1—figure 16.** Effects of $I_{KCa}$ enhancement on alternans generation in the chronic stage. Left: switching $I_{KCa}$ activity back to normal (black traces) caused AP prolongation and bigger alternans. Weaker $I_{KCa}$ also led to stronger CaT (bottom right, black solid line) and larger calcium release in the longer beat (black solid line) that was more difficult for calcium level to recover in JSR (upper right, black solid line).

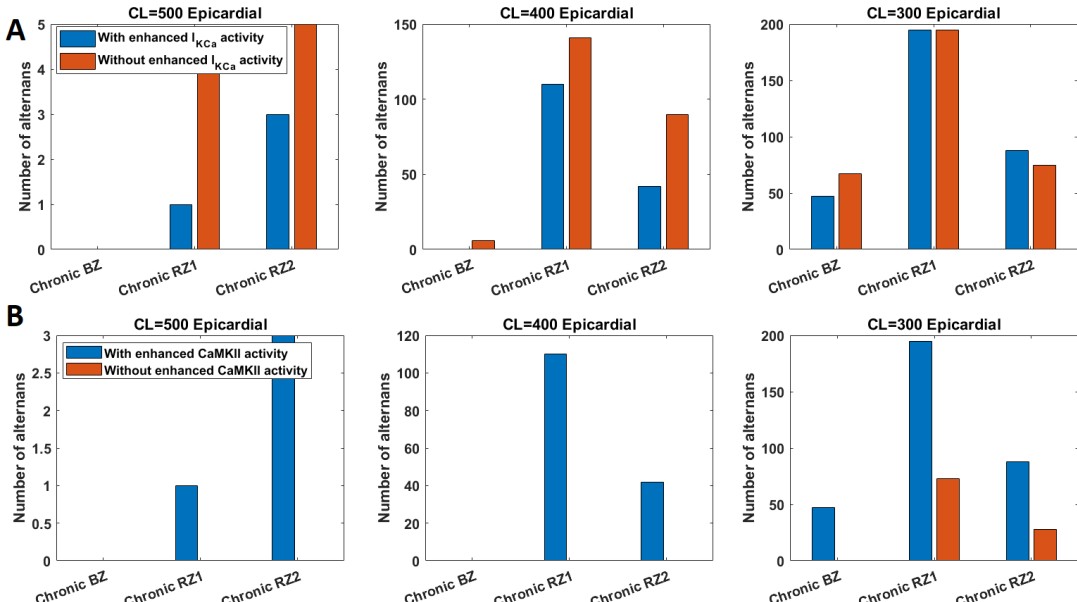

**Appendix 1—figure 17.** CaMKII and I$_{KCa}$ had opposite roles on alternans inducibility in chronic ionic current remodelling. Enhanced I$_{KCa}$ tended to inhibit alternans generation (**A**), whereas augmented CaMKII promoted alternans (**B**).

As stronger G$_{CaL}$ and stronger P$_{Jup}$ were consistently observed in the chronic population of alternans models (**Appendix 1—table 10**), the effects of reversing these trends were investigated. In a representative alternans model, when G$_{CaL}$ was inhibited by 20% (**Li et al., 2024**, the purple trace), the smaller calcium influx led to weaker calcium release, causing a milder reduction of JSR calcium level that was easier to refill, and leading to the elimination of alternans. If J$_{up}$ was further inhibited by 20% (**Appendix 1—figure 18**, the green trace), the slower calcium re-uptake led to lower initial JSR calcium level at the beginning of a beat, resulting in a smaller J$_{rel}$ and a milder JSR calcium reduction that was also easier to refill. However, G$_{CaL}$ and J$_{up}$ inhibition suppressed alternans generation at the cost of reducing CaT magnitude (**Appendix 1—figure 18**, CaT panel).

Although the chronic remodeling in the RZ decreased CaT amplitude compared with the NZ, the alternans models among the remodeling population can have relatively preserved CaT amplitudes (**Appendix 1—figure 19**). The relatively higher G$_{CaL}$ and P$_{Jup}$ in the alternans models contribute to bigger CaT$_{max}$ and smaller CaT$_{min}$ (**Appendix 1—table 10**, **Appendix 1—figure 19**). This suggests that models that were prone to alternans development may display a relative preserved CaT amplitude and LVEF.

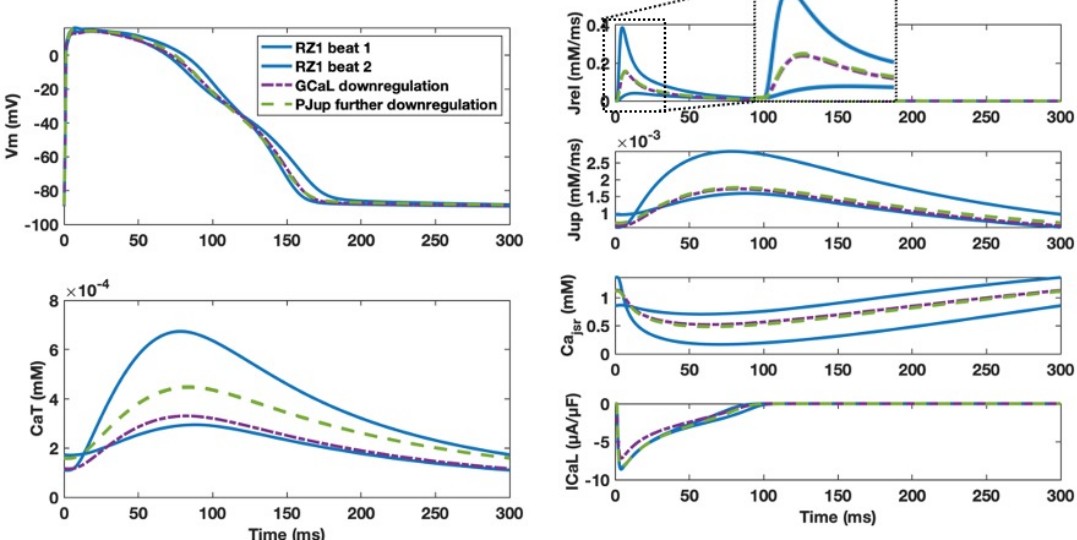

**Appendix 1—figure 18.** With chronic post-MI ionic current remodelling, alternans models needed stronger $G_{CaL}$ and more preserved $P_{Jup}$ to enable alternans generation.

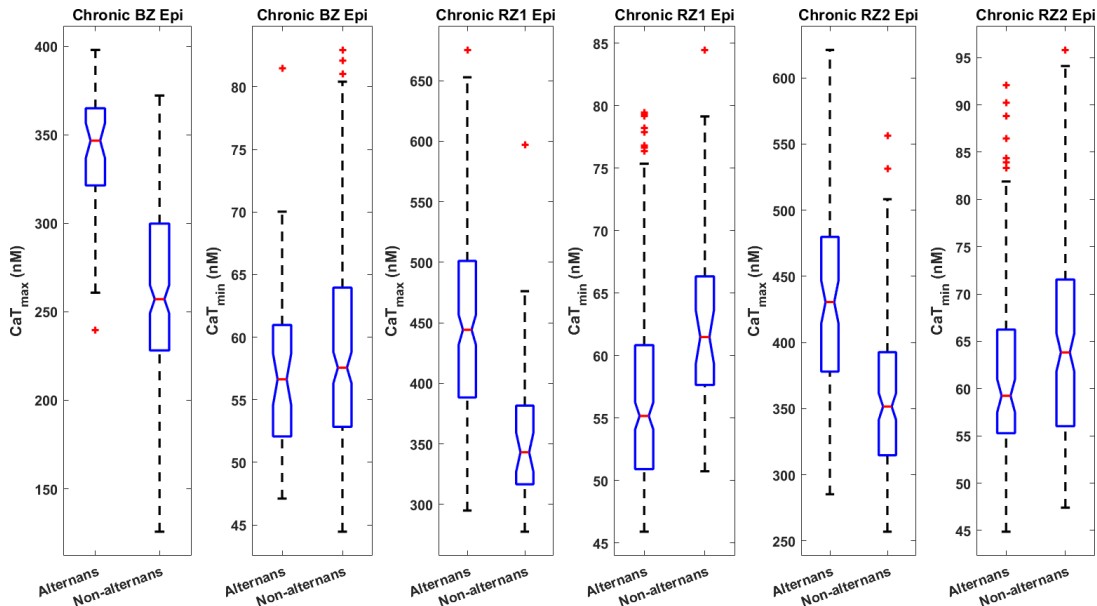

**Appendix 1—figure 19.** Alternans models had bigger $CaT_{max}$ than the non-alternating models in the epicardial populations with chronic post-MI remodelling (all with $P<0.001$). In addition, alternans models tended to have smaller $CaT_{min}$ in epicardial populations of Chronic RZ1 ($P<0.001$) and RZ2 ($P<0.05$), while the difference was not statistically significant for Chronic BZ.

## 11. Post-MI $I_{Kr}$ and $I_{NaL}$ remodeling promote EAD generations in models with preserved calcium magnitudes

All three types of chronic post-MI ionic remodeling promoted EADs and repolarization failure (RFs), and midmyocardium was most prone for the development of EAD (*Appendix 1—figure 20*, *Appendix 1—table 11*).

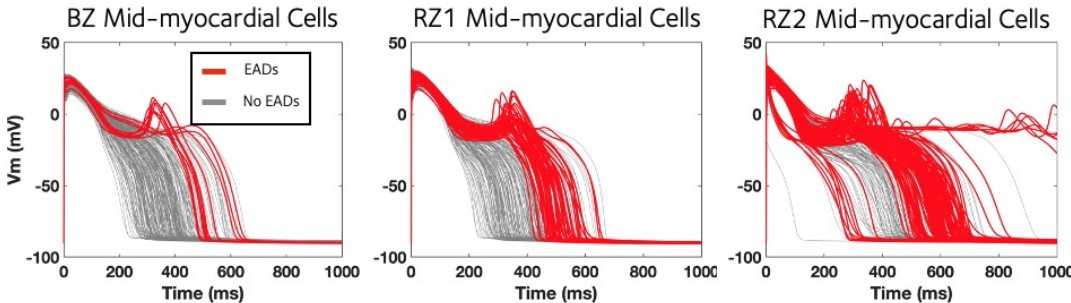

**Appendix 1—figure 20.** Midmyocardium was most prone for the development of EAD under chronic post-MI remodelling.

Cellular early afterdepolarizations (EADs) or repolarization failure (RF) were promoted in the post-MI population of chronic models, and we investigated the following two questions: (1) what are the key individual post-MI ionic remodeling contributing to the generation of EADs, and (2) what are the underlying ionic currents that determine whether a post-MI model is prone to EAD development?

To illustrate the effects of the individual chronic ionic remodeling on the inducibility of EADs, a representative model was chosen from the population (*Appendix 1—figure 21*), which had a normal AP at a CL of 1000ms in NZ (the blue trace). When the Chronic BZ ionic remodeling was introduced, an EAD was generated (the red trace). Removing the $I_{NaL}$ remodeling did not eliminate the EAD (the yellow trace), and similarly when $I_{Kr}$ inhibition was removed, EAD was still maintained (the purple trace). However, when both $I_{NaL}$ augmentation and $I_{Kr}$ inhibition were absent, the EAD was eliminated (the green trace). Although $I_{NaL}$ and $I_{Kr}$ remodeling were the key factors inducing EAD generation in the chronic stage, $I_{CaL}$ re-activation was also a necessary mechanism (the light blue trace).

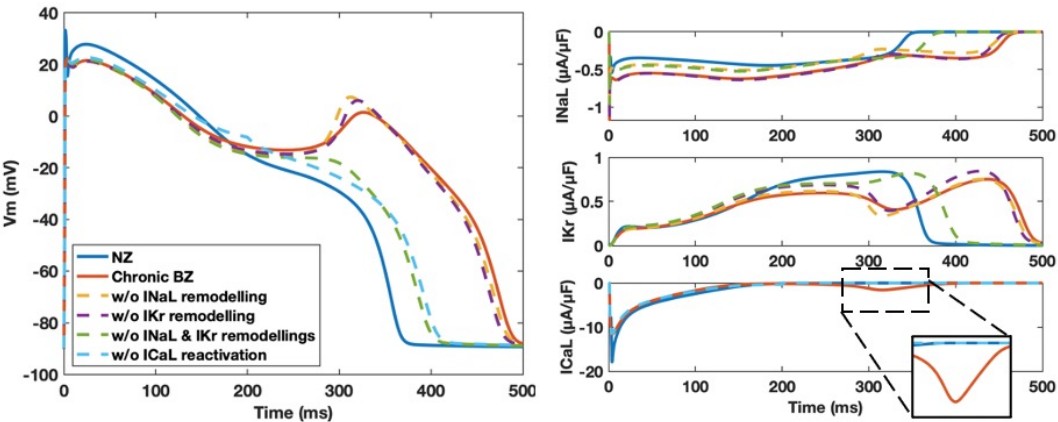

**Appendix 1—figure 21.** Chronic ionic current remodelling promotes EAD generation through the enhanced $I_{NaL}$ and suppressed $I_{Kr}$, which facilitate $I_{CaL}$ reactivation.

**Appendix 1—table 11.** Number of EADs and RFs induced by three chronic remodelling in endocardial, midmyocardial and epicardial population of models.

| Population of models (n=245) | No. of EADs and RFs at CL = 1000ms | Key parameters for EADs and RFs |
|---|---|---|
| Chronic BZ | Mid only (11) | ↑$G_{CaL}$, ↓$G_{Kr}$, ↑$G_{NCX}$ |
| Chronic RZ1 | Mid only (52) | ↑$G_{CaL}$, ↓$G_{Kr}$, ↑$G_{NCX}$ |
| Chronic RZ2 | Mid (118)>Epi (9)>Endo (1) | ↑$G_{CaL}$, ↓$G_{Kr}$, ↑$G_{NCX}$, ↑$P_{Jup}$ |

By comparing the parameters of EAD models against the non-EAD models in the chronic populations, stronger $G_{CaL}$, $G_{NCX}$ and weaker $G_{Kr}$ were consistently observed in the EAD populations (*Appendix 1—table 11*). Due to the stronger $G_{CaL}$ and $G_{NCX}$, the EAD models also displayed a stronger $CaT_{max}$ and a lower $CaT_{min}$ in all three ionic remodeling populations (*Appendix 1—figure*

*22*). Therefore, these results suggested that models which were prone to EAD development may present as a relative preserved LVEF.

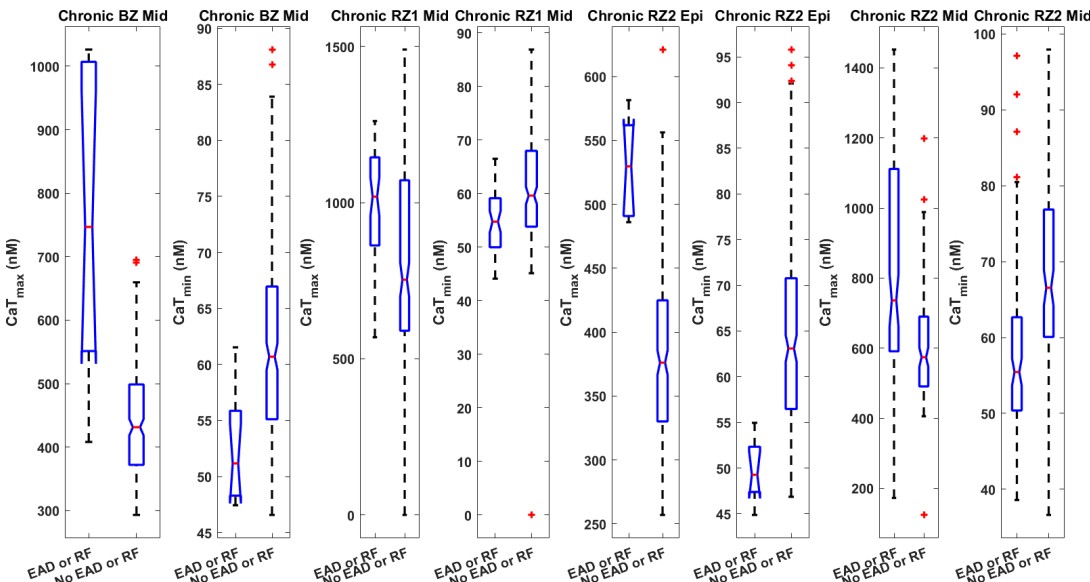

**Appendix 1—figure 22.** EAD models tended to have stronger CaT$_{max}$ and weaker CaT$_{min}$ in the population of Chronic BZ, RZ1 and RZ2 models (all with *P*<0.001).

## 12. Both sarcolemmal and calcium dynamics remodeling are necessary for the generation of post-MI EAD alternans

As shown in *Appendix 1—figure 14*, EADs can be a major cause of big alternans in the midmyocardial population of chronic post-MI models. Theoretically, calcium alternans can induce EADs under proper conditions, and EADs may display an alternating pattern which occurs in every other beat (*Qu and Weiss, 2023*). To explore whether the EAD alternans were predominantly EAD or alternans, the 10 representative EAD alternans examples in *Appendix 1—figure 14* were simulated without either EAD related remodelling (I$_{NaL}$ and I$_{Kr}$) or alternans related remodelling (P$_{Jup}$ and CaMKII). EAD alternans disappear when either of the two types of remodelling was switched off (*Appendix 1— figures 23 and 24*), indicating both the sarcolemmal ionic current remodelling and the altered calcium dynamics are necessary conditions for the generation of these EAD alternans.

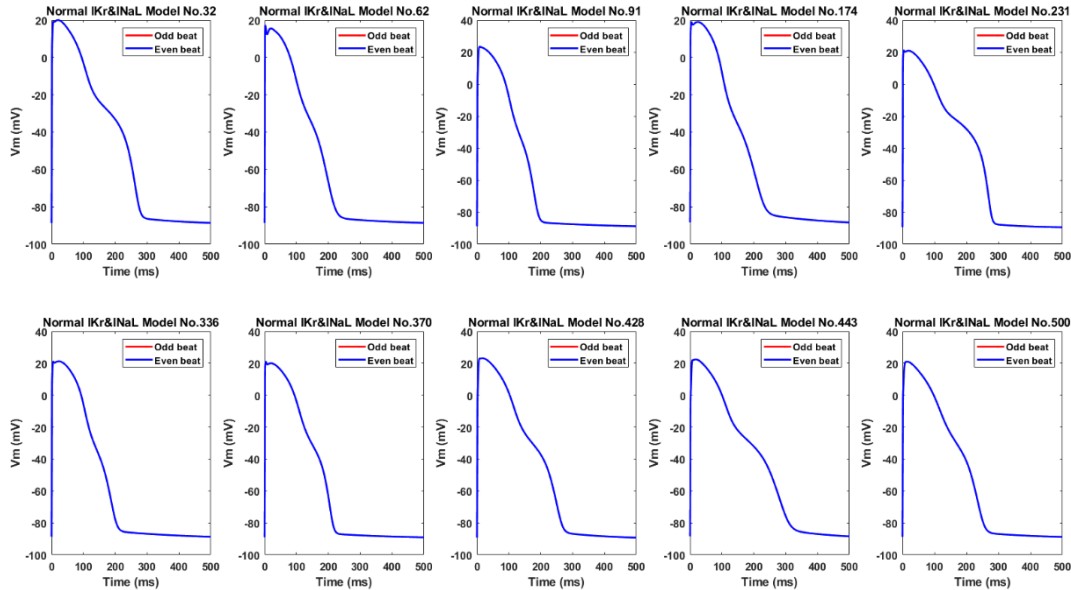

**Appendix 1—figure 23.** Chronic BZ remodelling induced EAD alternans in the ten representative midmyocardial models, but when these models were simulated without $I_{Kr}$ and $I_{NaL}$ remodelling, neither EAD nor alternans occurred.

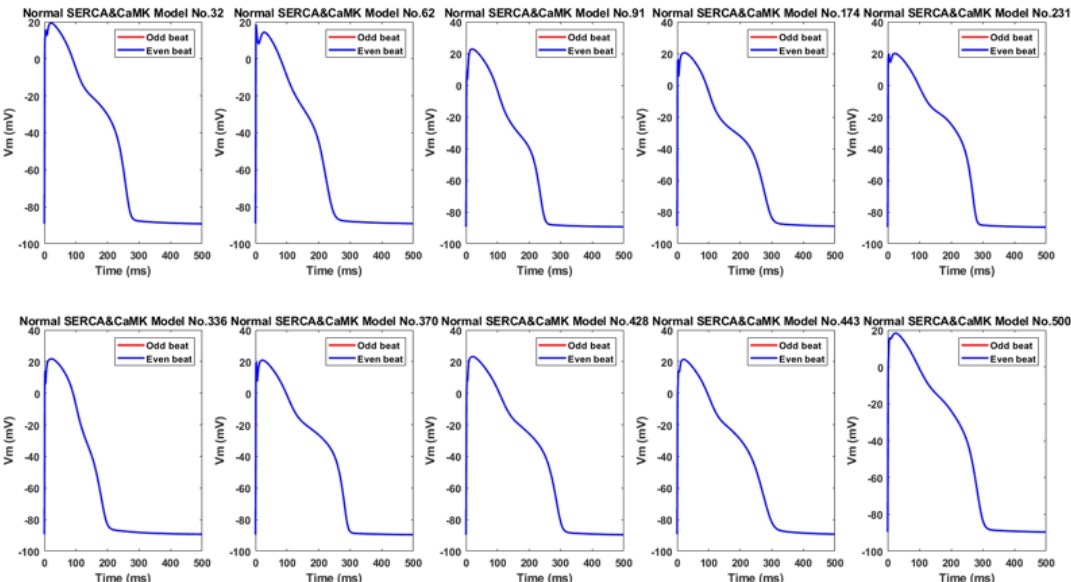

**Appendix 1—figure 24.** Chronic BZ remodelling induced EAD alternans in the ten representative midmyocardial models, but when these models were simulated without $P_{Jup}$ (SERCA) and CaMKII remodelling, neither EAD nor alternans occurred.

