## [Editor Report · eLife Assessment]

This computational study integrates detailed electrophysiology and mechanical contraction predictions, which are often modeled separately. The findings of this **important** work are that abnormal ECGs that are associated with higher risk of sudden cardiac death are predicted to have almost no relationship with left ventricular ejection fraction, which is conventionally used as a risk factor for arrhythmia. The conclusions are based on **compelling** evidence for the need of incorporating additional risk factors for assessing post-myocardial infarction patients.

---

## [Referee Report · Reviewer #1 (Public review)]

Summary:

In this study from Zhou, Wang, and colleagues, the authors utilize biventricular electromechanical simulations to illustrate how different degrees of ionic remodeling can contribute to different ECG morphologies that are observed in either acute or chronic post-myocardial infarction (MI) patients. Interestingly, the simulations show that abnormal ECG phenotypes - associated with higher risk of sudden cardiac death - are predicted to have almost no correspondence with left ventricular ejection fraction, which is conventionally used as a risk factor for arrhythmia.

Strengths:

The numerical simulations are state-of-the-art, integrating detailed electrophysiology and mechanical contraction predictions, which are often modeled separately. The population of ventricular simulations provide mechanistic interpretation, down to the level of single cell ionic current remodeling, for different types of ECG morphologies observed in post-MI patients. Collectively, these results demonstrate compelling and significant evidence for the need of incorporating additional risk factors for assessing post-MI patients.

The authors have addressed all of my previous concerns in this updated version.

---

## [Referee Report · Reviewer #2 (Public review)]

Summary:

The authors constructed a multi-scale modeling and simulation methods to investigate the electrical and mechanical properties under acute and chronic myocardial infarction (MI). The simulated three acute MI conditions and two chronic MI conditions. They showed that these conditions gave rise to distinct ECG characteristics that have seen in clinical settings. They showed that the post-MI remodeling reduced ejection fraction up to 10% due to weaker calcium current or SR calcium uptake, but the reduction of ejection fraction is not sensitive to remodeling of the repolarization heterogeneities.

Strengths:

The major strength of this study is the construction of the computer modeling that simulates both electrical behavior and mechanical behavior for post-MI remodeling. The links of different heterogeneities due to MI remodeling to different ECG characteristics provide some useful information for understanding the complex clinical problems.

Weaknesses:

The rationale (e.g., physiological or medical bases) for choosing the 3 acute MI and 2 chronic MI settings is not clear. Although the authors presented a huge number of simulation data, in particular in the supplemental materials, it is not clearly stated what novel findings or mechanistic insights that this study gained beyond the current understanding of the problem.

---

## [Author Response]

The following is the authors’ response to the original reviews.

**Public Reviews:**

**Reviewer #1 (Public Review):**
Summary:In this study by Zhou, Wang, and colleagues, the authors utilize biventricular electromechanical simulations to illustrate how different degrees of ionic remodeling can contribute to different ECG morphologies that are observed in either acute or chronic post-myocardial infarction (MI) patients. Interestingly, the simulations show that abnormal ECG phenotypes - associated with a higher risk of sudden cardiac death - are predicted to have almost no correspondence with left ventricular ejection fraction, which is conventionally used as a risk factor for arrhythmia.Strengths:The numerical simulations are state-of-the-art, integrating detailed electrophysiology and mechanical contraction predictions, which are often modeled separately. The simulation provides mechanistic interpretation, down to the level of single-cell ionic current remodeling, for different types of ECG morphologies observed in post-MI patients. Collectively, these results demonstrate compelling and significant evidence for the need to incorporate additional risk factors for assessing post-MI patients.Weaknesses:The study is rigorous and well-performed. However, some aspects of the methodology could be clearer, and the authors could also address some aspects of the robustness of the results. Specifically, does variability in ionic currents inherent in different patients, or the location/size of the infarct and surrounding remodeled tissue impact the presentation of these ECG morphologies?

We thank the reviewer for their considered evaluation. In response to the reviewer’s comments regarding variability in ionic currents, we have added simulations using a n=17 populations of models with variability in ionic conductances in the baseline ToR-ORd model to the paper, to show the effect of such variation on the post-MI ECG presentation in acute and chronic conditions. This is now described in the Methods [lines 140, 158-161, 242-244, 245-246, 261-263], and shown in the methods Figure 1A, 1B. The ECG results using this population of models are shown in Figure 2C and described in [lines 333-335] and the pressure volume results using the population of models are shown in Figure 5A and 5B and described in [lines 417-418, 442-444, 448-450]. The population of models showed consistent patterns in both the ECG and LVEF as the baseline model, this is discussed in [lines 563-564, 688-690].

Regarding the effect of scar location and size on the ECG, we refer the reader and reviewer to a related paper where this is explored in depth using a formal sensitivity analysis and deep learning inference (https://pubmed.ncbi.nlm.nih.gov/38373128/). This is better able to do justice to this question rather than overloading this paper with additional investigations. We include a reference to this paper in the discussion section [lines 694-695].

**Reviewer #2 (Public Review):**
Summary:The authors constructed multi-scale modeling and simulation methods to investigate the electrical and mechanical properties of acute and chronic myocardial infarction (MI). They simulated three acute MI conditions and two chronic MI conditions. They showed that these conditions gave rise to distinct ECG characteristics that have been seen in clinical settings. They showed that the post-MI remodeling reduced ejection fraction up to 10% due to weaker calcium current or SR calcium uptake, but the reduction of ejection fraction is not sensitive to remodeling of the repolarization heterogeneities.Strengths:The major strength of this study is the construction of computer modeling that simulates both electrical behavior and mechanical behavior for post-MI remodeling. The links of different heterogeneities due to MI remodeling to different ECG characteristics provide some useful information for understanding complex clinical problems.Weaknesses:The rationale (e.g., physiological or medical bases) for choosing the 3 acute MI and 2 chronic MI settings is not clear. Although the authors presented a huge number of simulation data, in particular in the supplemental materials, it is not clearly stated what novel findings or mechanistic insights this study gained beyond the current understanding of the problem.

We thank the reviewer for their careful evaluations of our work. The justification for selecting the 3 acute MI and 2 chronic MI states is based on clinical and experimental reports, as summarised in the Methods section [lines 245-247, 252-256, 264-266]. We have also highlighted the key novelty and significance of the study in the Discussion [lines 579-582].

**Recommendations for the authors:**

**Reviewer #1 (Recommendations For The Authors):**
(1) This was clarified very late in the Discussion, but for most of the paper, I was unclear if heart geometry was the same for all simulations. Presumably, this includes the size and location of the infarct, BZ, and RZ. It would be helpful to clarify this in the Methods.

This has been clarified in the first paragraph of the Methods section [lines 142-145].

(2) On lines 224-226, the Methods refers to implementing several population members from the ToR-ORd model (in addition to the baseline) into the biventricular EM simulations. Is this in reference to the simulations shown in Figures 6 and 7, or different simulations? Please clarify.

We now randomly select 17 of the 245 cell models in the population to be embedded in ventricular simulations, to produce a ventricular population of models. This allows us to explore the effect that physiological variability in the baseline ionic conductances has on the phenotypic representation of ionic remodellings in the ECG and LVEF. An explanation of this can be found in the Methods section [lines 241-244].

For Figures 6 and 7, we selected two arrhythmic cell models from the n=245 population of cell models to be embedded into two ventricular simulations to demonstrate the arrhythmic potential of the cellular model at ventricular scale. This has been clarified in Methods [lines 269-271].

Additionally, for the cases where a population member is used, are all regions of the ventricles "scaled" in the same manner, or were only the properties of the particular region drawn from the population modified relative to baseline (e.g., mid-myocardial cells in Figure 6)?

The cells were embedded according to transmural heterogeneity in the remote zone for Figures 6 and 7. This has been clarified in the Methods [line 271-273].

(3) Interestingly, the study finds that the ionic remodeling in different peri-infarct regions to be most critical in the ECG phenotype, which at least strongly suggests that inherent intra-patient variability in ion channel expression could also be critical.This is related to the comment on the use of population members. If the authors utilized one of the ventricular myocyte population members as the 'reference' (instead of the baseline ToR-ORd parameters) and applied the same types of remodeling as in Figures 3 and 4, would they expect the same ECG morphologies?

We have now performed this test and selected 17 cell models from the population to create a ventricular population of models. On top of this ventricular population, we have applied the remodellings, and showed that the simulated ECG morphologies were mostly consistent across these 20 members (Figure 2C).

(4) Related, do the authors expect that the location and/or size of the infarct and peri-infarct regions would impact the different ECG morphologies?

Regarding the effect of scar location and size on the ECG, we refer the reader and reviewer to a related paper where this is explored in depth using a formal sensitivity analysis and deep learning inference (https://pubmed.ncbi.nlm.nih.gov/38373128/). We feel this is better able to do justice to this question rather than overloading this paper with additional investigations. We include a reference to this paper in the discussion section [lines 694-695].

**Reviewer #2 (Recommendations For The Authors):**
(1) Although the authors listed the parameters and cited the papers for the origins of the parameter changes in SM4 and table S4, it should be summarized in the methods section what are the major changes or differences for the 5 conditions. Furthermore, it should be stated what is the rationale for choosing these conditions. Are these choices based on clinical classifications or experimental conditions?

The major differences between the 5 conditions have now been summarised in the Methods [lines 252-256, 264-266]. These remodellings have been collated from a range of experimental measurements in both human and animal data, which are summarised in Table S4. This has been clarified in Methods [lines 245-247].

(2) Figure 3C and Figure 4C do not add any additional information beyond the conductance changes listed in Table 4, and I'd suggest removing them from the figures. On the other hand, it took me some time to look at Table 4 to figure out the corresponding changes. As commented above, the remodeling changes should be summarized in the main text to help reading.

Figure 3C and 4C provide a visual explanation of the ionic remodellings in these conditions to echo the added descriptions in the text [lines 252-256, 264-266]. For this reason, we have elected to keep those figures in the manuscript.

(3) The authors presented a large amount of data in Supplemental Materials, some may be unnecessary and some are difficult to follow. For example; (1) There is a lot of data in Table S6, there is a simple mention in the main text and Table S6 legend. A summary of the data is needed for the readers to understand the properties of the different conditions, instead of letting the readers figure them out from the table. The same should be done for other tables and figures. There are some format issues for the tables, which mess up some of the numbers and text. (2) The data shown in Figures S25-29 provide almost no new information beyond the well-known effects of ionic currents on EAD genesis, i.e., EADs are promoted by inward currents and suppressed by outward currents. The data for alternans (Figures S18-22) are a little more complex than the cases for EADs, I think that they can be simplified.

Thanks for the suggestions. We have now extracted the key information from Table S6- S9 and summarized them in the caption. We have also fixed the layout of the tables in this revision. The supplementary sections on alternans and EADs are simplified with the key parameters related to these proarrhythmic phenomena summarized in tables instead of showing all boxplots of parameter distributions (Tables S10 and S11).

(4) The authors showed two mechanisms of alternans: EAD-driven and Ca-driven alternans in chronic MI. There are several distinct mechanisms of alternans including EAD-induced alternans (see the recent review by Qu and Weiss, Circ Res 132, 127(2023)). Theoretically, calcium alternans can also induce EAD alternans under proper conditions, can you rule out that the EAD alternans are not due to Ca alternans? The results in Fig.7D may say the opposite. There are some chicken-or-egg issues here.

In Figure 7D, we showed that the epicardial cell type (blue trace) had stable EADs at fast pacing with no calcium alternans, while both the endocardial (red trace) and mid-myocardial (green trace) cell types failed to fully repolarise in every other beat. To explore whether the EAD alternans are driven by calcium alternans, we tested the effects of switching off the alternans related remodelling, and the APs tuned out to be normal. On the other hand, when we turned off the EAD related remodelling, neither EADs nor alternans occurred. Therefore, the results show the two types of ionic current remodelling are both necessary for the generation of EAD alternans (lines 656-659 in the discussion and SM9).

(5) As for the formation of ectopic beats, it can be caused by EADs but it can caused by repolarization gradient, they are not the same and differ in different AP models (Liu et al, CircAE 12, e007571 (2019), Zhang et al, Biophy J 120, 352(2021)). It is not clear here whether the primary cause is repolarization gradient or EADs. At tissue, EADs tend to be suppressed by repolarization gradient, there is a goldilocks between the EAD amplitude and repolarization gradient for an ectopic beat to form.

When isolated cells that showed EAD were embedded in ventricular tissue, we saw ectopic wave propagation. This was because the EADs in the RZ generated conduction block, which enabled a large repolarisation gradient to form between the BZ and RZ, thereby leading to ectopy. This has been clarified in the Results [lines 507-510].

Additionally, we have clarified the presence of the EADs in the ventricular simulations by labelling where this occurs in the green, purple, and yellow traces in Figure 7C. This was easily missed before due to the stretched proportions of the traces in the x-axis, which is necessary to show clearly the repolarisation gradients that drive ectopy.

(6) The authors showed many population simulations. I guess that they are all in single cells. If the population simulations were done in the whole heart, it should be stated how many models were simulated. If only one of the population models was selected for the whole heart for each case, it should clarify the rationale for choosing one of the many models. If populations of cells were modeled in the whole heart, clarify how the models were distributed in the heart.

We now randomly select 17 of the 245 cell models in the population to be embedded in ventricular simulations, to produce a ventricular population of models. This allows us to explore the effect that physiological variability in the baseline ionic conductances has on the phenotypic representation of ionic remodellings in the ECG and LVEF. An explanation of this can be found in the Methods section [lines 241-244]. Whenever the cell models are embedded in the relevant zones, they are uniformly distributed according to the transmural heterogeneity [lines 271-273].

(7) QRS intervals in the simulations are much wider than the real recordings from patients (Figure 2 and Table S8). At least, a QRS of 120 ms for normal control is too wide and probably not normal.

We have manually measured QRS duration and updated the delineation method to calculate the other biomarkers. The new values now lie within normal ranges and have been updated in SM Table S7 and S8 and in Figure 2, and the new delineation method has been included in SM2.